# A CD4-mimetic compound enhances vaccine efficacy against stringent immunodeficiency virus challenge

Navid Madani[1,2,3], Amy M. Princiotto[1], Linh Mach[4], Shilei Ding[5,6], Jérémie Prevost[5,6], Jonathan Richard[5,6], Bhavna Hora[7], Laura Sutherland[7], Connie A. Zhao[1], Brandon P. Conn[4], Todd Bradley[7], M. Anthony Moody[7], Bruno Melillo[8], Andrés Finzi[5,6], Barton F. Haynes[7], Amos B. Smith III[8], Sampa Santra[4] & Joseph Sodroski[1,2,9]

The envelope glycoprotein (Env) trimer ((gp120/gp41)$_3$) mediates human immunodeficiency virus (HIV-1) entry into cells. The "closed," antibody-resistant Env trimer is driven to more open conformations by binding the host receptor, CD4. Broadly neutralizing antibodies that recognize conserved elements of the closed Env are potentially protective, but are elicited inefficiently. HIV-1 has evolved multiple mechanisms to evade readily elicited antibodies against more open Env conformations. Small-molecule CD4-mimetic compounds (CD4mc) bind the HIV-1 gp120 Env and promote conformational changes similar to those induced by CD4, exposing conserved Env elements to antibodies. Here, we show that a CD4mc synergizes with antibodies elicited by monomeric HIV-1 gp120 to protect monkeys from multiple high-dose intrarectal challenges with a heterologous simian-human immunodeficiency virus (SHIV). The protective immune response persists for at least six months after vaccination. CD4mc should increase the protective efficacy of any HIV-1 Env vaccine that elicits antibodies against CD4-induced conformations of Env.

---

[1] Department of Cancer Immunology and Virology, Dana-Farber Cancer Institute, Boston, MA 02215, USA. [2] Department of Microbiology and Immunobiology, Harvard Medical School, Boston, MA 02115, USA. [3] Department of Global Health and Social Medicine, Harvard Medical School, Boston, MA 02115, USA. [4] Center for Virology and Vaccine Research, Beth Israel Deaconess Medical Center, Harvard Medical School, Boston, MA 02215, USA. [5] Centre de Recherche du CHUM, Montreal, QC H2X 0A9, Canada. [6] Department of Microbiology, Infectious Diseases and Immunology, Université de Montréal, Montreal, QC H2X 0A9, Canada. [7] Department of Medicine, Department of Immunology, Duke Human Vaccine Institute, Duke University Medical Center, Durham, NC 27710, USA. [8] Department of Chemistry, University of Pennsylvania, Philadelphia, PA 19104, USA. [9] Department of Immunology and Infectious Diseases, Harvard T.H. Chan School of Public Health, Boston, MA 02115, USA. Correspondence and requests for materials should be addressed to N.M. (email: navid_madani@dfci.harvard.edu) or to A.B.S.III. (email: smithab@sas.upenn.edu) or to J.S. (email: joseph_sodroski@dfci.harvard.edu)

The envelope glycoprotein (Env) trimer, consisting of three gp120 exterior glycoproteins and three gp41 transmembrane glycoproteins, mediates the entry of human immunodeficiency virus (HIV-1) into cells[1]. The "closed", pre-triggered conformation of Env is driven into more "open" conformations by binding of the virus to the host receptor, CD4[2,3]. Env is the only HIV-1-specific target for antibodies that mediate virus neutralization and antibody-dependent cell cytotoxicity (ADCC), both of which can potentially contribute to protection from HIV-1 infection[1,4–12]. As a successful persistent virus, HIV-1 has evolved Env characteristics and regulatory proteins that minimize the elicitation and effectiveness of host antibodies[1,10,13–20]. Most broadly neutralizing antibodies bind conserved epitopes on the closed Env trimer[21,22]; however, broadly neutralizing antibodies are elicited inefficiently during natural HIV-1 infection[23,24], and immunization strategies that consistently elicit broadly neutralizing antibodies in monkeys or humans have not yet been developed[13,14]. By contrast, poorly neutralizing antibodies that recognize more open Env conformations are elicited at high titers during natural HIV-1 infection and by vaccination[25–27]. However, the binding of these antibodies to the Env–CD4 complex in the virus-target cell synapse is sterically restricted, accounting for their minimal virus-neutralizing potential[15]. Some readily elicited antibodies mediate ADCC against HIV-1-infected cells by recognizing highly conserved gp120 "Cluster A" epitopes that are exposed upon CD4 binding[8–10]. Downregulation of cell-surface Env–CD4 complexes by the HIV-1 Vpu and Nef proteins contributes to virus escape from host ADCC responses[8,16–20].

CD4-mimetic compounds (CD4mc) are small molecules (MW < 600 Da) that bind HIV-1 gp120 within the well-conserved Phe 43 cavity, near the binding site for CD4[28–30]. Rational, structure-based design has improved the potency and breadth of CD4mc[31,32]. The binding of CD4mc induces conformational changes in Env similar to those induced by CD4[33,34]. However, in contrast to CD4-bound Env states[8,16–20], the Env conformations induced by CD4mc on viruses and infected cells are readily accessible to antibodies[20,34–38]. Thus, CD4mc nullify many of the strategies that HIV-1 has evolved to evade the host antibody response.

Here, we test whether vaccination with HIV-1 gp120 immunogens can confer protection against a stringent challenge with a heterologous simian-human immunodeficiency virus (SHIV-C5)[39] that has been sensitized by a CD4mc, BNM-III-170[31] (Fig. 1).

## Results

**Immunization of monkeys with HIV-1 gp120 Envs.** Three groups of rhesus macaques were studied (Supplementary Table 1 and Supplementary Fig. 1). Group 1 monkeys were immunized with human serum albumin (HSA) and were challenged with SHIV-C5 mixed with the CD4mc, BNM-III-170. Group 2 monkeys were immunized with HIV-1$_{CH505}$ gp120 variants and challenged with SHIV-C5 mixed with the DMSO vehicle. Group 3 monkeys were immunized with HIV-1$_{CH505}$ gp120 variants and challenged with SHIV-C5 mixed with BNM-III-170.

The titers of anti-Env antibodies in the sera of the Group 2 and Group 3 monkeys immunized with HIV-1$_{CH505}$ gp120 were comparable (Supplementary Fig. 2). These sera did not efficiently neutralize primary HIV-1 in the absence of the CD4mc, but inhibited the infection of heterologous primary viruses that were sensitized by incubation with sub-neutralizing concentrations of BNM-III-170 (Fig. 2 and Supplementary Table 2). Apparently, the antibodies that mediate the neutralization of BNM-III-170-sensitized viruses recognize Env elements that are conserved between phylogenetic Clade C and Clade B HIV-1. Of note, the sera from all monkeys in Group 2 and Group 3 efficiently inhibited the infection of viruses with the HIV-1$_{C5}$ Env (corresponding to that of the challenge SHIV-C5) only when BNM-III-170 was present. Thus, immunization of monkeys with Clade C HIV-1 gp120 variants elicited antibodies capable of neutralizing diverse CD4mc-sensitized primary HIV-1, including a heterologous Clade C transmitted/founder virus.

Sera from the monkeys in Group 2 and Group 3, but not Group 1, detectably bound the HIV-1$_{C5}$ Env on the surface of SHIV-C5-infected peripheral blood CD4$^+$ T lymphocytes, and this binding was increased after incubation with BNM-III-170 (Fig. 3a). Sera from the Group 2 and Group 3 monkeys, but not the Group 1 animals, efficiently mediated ADCC against SHIV-C5-infected CD4$^+$ T lymphocytes in the presence of BNM-III-170 (Fig. 3b). This ADCC activity was decreased by the addition of the Fab fragment of the A32 antibody, which recognizes a "Cluster A" epitope on HIV-1 gp120[10,38] (Fig. 3c). These results indicate that a substantial amount of the observed ADCC activity is dependent on antibody access to Cluster A epitopes on gp120.

**SHIV-C5 challenges of immunized monkeys.** The monkeys in Groups 1, 2, and 3 were challenged three times intrarectally with a high dose (~3.5 animal infectious dose (AID$_{50}$) units) of SHIV-C5. Each of the three challenges was conducted 2–4 weeks after boosting with HSA (Group 1) or gp120 (Groups 2 and 3) (Fig. 4a). SHIV-C5 contains the Env derived from the Clade C transmitted/founder HIV-1$_{C5}$ and is a relatively neutralization-resistant (Tier 2/3) virus[39]. The Envs of the SHIV-C5 challenge virus and the HIV-1$_{CH505}$ strain from which the gp120 immunogens were derived exhibit ~81% amino acid identity; this level of sequence divergence is comparable to that exhibited by Clade C HIV-1 Envs in general (Supplementary Table 3 and Supplementary Fig. 3)[40]. The Group 1 and Group 3 monkeys were challenged with SHIV-C5 that was mixed with BNM-III-170 (final concentration 300 μM) within 30 min of intrarectal inoculation. The Group 2 monkeys were challenged with SHIV-C5 that was mixed with the DMSO vehicle only. Plasma viral RNA levels in the challenged monkeys were monitored weekly, and monkeys that were viremic on at least two occasions were considered infected.

**Fig. 1** Structure of the CD4mc, BNM-III-170, used in this study. The trifluoroacetate (TFA) salt of BNM-III-170 was synthesized from 5-bromo-1-indanone following the procedures in ref. [31]

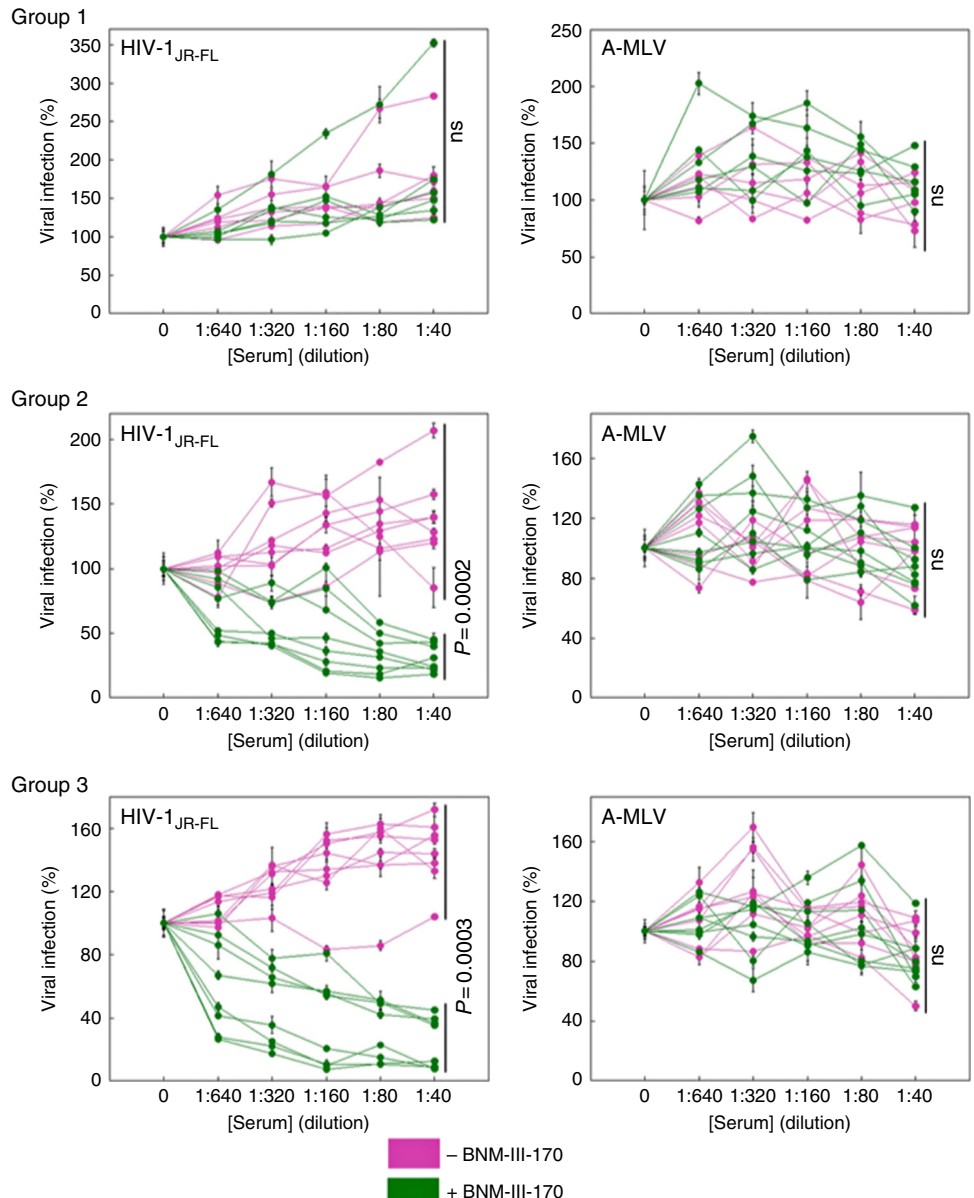

**Fig. 2** CD4mc-dependent HIV-1 neutralization by immunized monkey sera. Monkeys were immunized with either human serum albumin (HSA) (Group 1) or the Clade C HIV-1$_{CH505}$ gp120 (Groups 2 and 3). In the experiments shown in the green curves, recombinant HIV-1 encoding luciferase and pseudotyped with the heterologous Clade B HIV-1$_{JR-FL}$ Env (left panels) or with the control amphotropic murine leukemia virus (A-MLV) Env (right panels) was incubated with 20 μM BNM-III-170 for 30 min at 37 °C; in the experiments shown in the magenta curves, the viruses were incubated with the equivalent volume of DMSO under the same conditions. The viruses were then incubated for 30 min at 37 °C with the indicated dilution of sera from the immunized monkeys (collected on the day of SHIV-C5 Challenge 1). The virus-serum mixtures were added to Cf2Th-CD4/CCR5 cells and, 48 h after continuous incubation at 37 °C, the cells were lysed and luciferase activity measured. The percentage of luciferase activity (which reflects the level of virus infection) relative to that seen for viruses without added serum is shown. The means and standard deviations of values obtained from triplicate assays are shown. Differences between the sets of curves obtained in the absence and presence of BNM-III-170 were evaluated for statistical significance using the statmod software package for R (http://bioinf.wehi.edu.au/software/compareCurves). (ns not significant)

The Kaplan–Meier analysis[41] of the percentage of monkeys that remained uninfected after SHIV-C5 challenge is shown in Fig. 4b. In total 7 of the 8 monkeys in Group 2 became infected after one inoculation of SHIV-C5 and the remaining uninfected monkey became infected on the subsequent challenge. The susceptibility of the Group 2 monkeys to this SHIV-C5 challenge was similar to that expected for unvaccinated rhesus macaques[39,42] (Fig. 4c). The rate of SHIV-C5 acquisition was lower for the Group 1 monkeys than for the Group 2 animals, perhaps as a result of the direct antiviral effects of the CD4mc[28–32]; however,

this difference was not significant ($P = 0.32$, log-rank test). After three SHIV-C5 challenges, 5 out of 6 monkeys in Group 1 became infected. In Group 3, all of the monkeys resisted the first two challenges, and 6 out of 8 monkeys resisted the third challenge. Compared with the Group 2 monkeys that received the gp120 vaccine alone, the monkeys in Group 3 that received the gp120 vaccine and CD4mc were significantly protected from SHIV-C5 infection ($P = 0.0039$, log-rank test). The Group 3 monkeys were also significantly protected from SHIV-C5 infection compared with the monkeys in Group 1 that were immunized with human

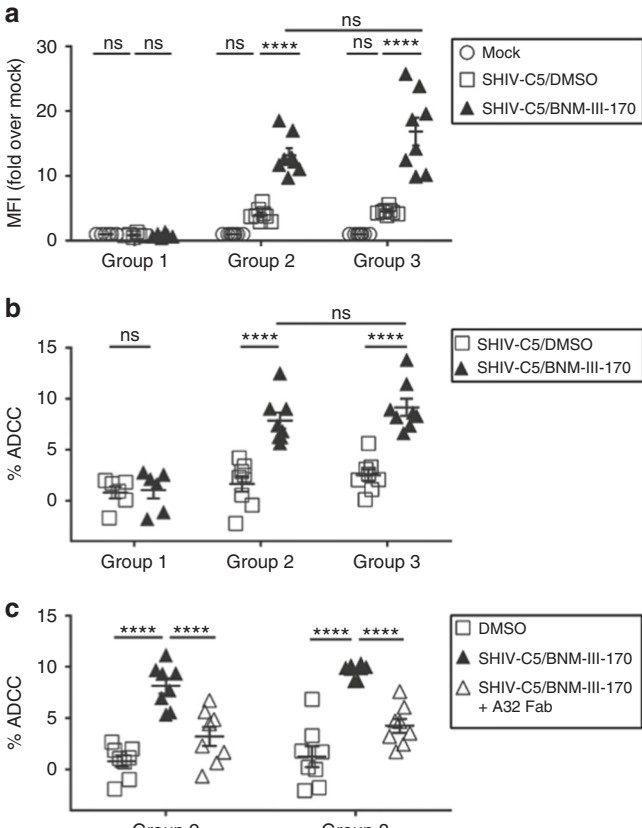

**Fig. 3** CD4mc-dependent ADCC killing by immunized monkey sera. Primary CD4[+] T lymphocytes from three healthy HIV-1-negative individuals were activated[37], and then infected with SHIV-C5 virus for 48 h before performing the antibody binding and ADCC assays. **a** Surface staining of infected cells was done with 1:250 dilutions of sera collected from the monkeys on the day of SHIV-C5 Challenge 1. Serum was added in the presence of 50 μM BNM-III-170 or an equivalent volume of DMSO (negative control) at 37 °C for 1 h. Goat anti-human antibody (AF-647) was used as the secondary antibody. The Mean Fluorescence Intensity (MFI) of AF-647 staining on Aqua Vivid (Invitrogen)-negative cells (living cells) is shown, relative to that of mock-infected cells. See Supplementary Fig. 9 for an example of the FACS gating strategy. **b** To evaluate ADCC, SHIV-C5-infected primary CD4[+] T lymphocytes were incubated with autologous peripheral blood mononuclear cells (Effector:Target ratio of 10:1) from the same donor for 4–6 h at 37 °C in the presence of a 1:250 dilution of monkey serum and 50 μM BNM-III-170 or equivalent volume of DMSO. The serum was collected from the monkeys on the day of SHIV-C5 Challenge 1. Infected cells were identified by staining with an Alexa Fluor (AF)-488-conjugated antibody against SIV p27 CA. The percentage of ADCC killing was determined with the following formula: [(% of p27 + cells in Targets plus Effectors)−(% of p27 + cells in Targets plus Effectors plus serum)]/(% of p27 + cells in Targets) by gating infected (p27[+]) live (Aqua Vivid-negative) target cells, as described[37]. **c** To evaluate the effect of the A32 Fab on the susceptibility of SHIV-C5-infected cells to ADCC mediated by the serum samples, 5 μg/ml of A32 Fab was added to the ADCC assay described in **b**. The results shown represent the average of three experiments. Statistical significance was tested by One-way ANOVA. (****$P < 0.0001$; ns, not significant)

serum albumin and received the CD4mc ($P = 0.020$, log-rank test). These results suggest that the CD4mc, BNM-III-170, synergized with a gp120-elicited immune response to protect monkeys from high-dose intrarectal challenges with a heterologous Clade C transmitted/founder SHIV.

To examine the durability of the protective response, the uninfected monkeys in Groups 1 and 3 were challenged with SHIV-C5 mixed with BNM-III-170 at 17 weeks (Challenge 4) and 24 weeks (Challenge 5) after the last gp120 immunization. Following Challenge 4 at 17 weeks after immunization, the sole uninfected monkey in Group 1 became infected, whereas 4 out of 6 monkeys in Group 3 resisted the SHIV-C5 challenge (Fig. 4d). Following Challenge 5 at 24 weeks after immunization, 4 out of 4 monkeys in Group 3 resisted the SHIV-C5 challenge. These results indicate that the protective responses persist for at least six months after immunization. At 33 weeks after the last gp120 boost, the four uninfected monkeys in Group 3 were challenged with SHIV-C5 in the absence of BNM-III-170 (Challenge 6). All four monkeys became infected, underscoring the contribution of the CD4mc to the observed protection. The levels of viremia were comparable in the infected monkeys (Supplementary Figs. 4 and 5).

**Characterization of the infecting viruses.** To investigate the properties of the infecting viruses, we used single-genome amplification to obtain the env sequences of the SHIV-C5 challenge stock and the circulating viruses in the infected monkeys at the time of peak viremia. A moderate degree of Env sequence variation was documented in the SHIV-C5 challenge stock and in the circulating viruses from different monkeys (Fig. 5 and Supplementary Fig. 6a). The Envs of the infecting viruses were related to multiple variants detected in the SHIV-C5 challenge stock, with no consistent differences among the viruses infecting the three monkey groups. The infecting virus Envs from the Group 3 monkeys remained susceptible to direct inhibition by BNM-III-170 and, after incubation with BNM-III-170, to neutralization by serum from the monkeys at the time of virus challenge (Supplementary Fig. 6b). These results are consistent with a stochastically determined process of virus transmission in this SHIV-C5 model.

## Discussion

In these proof-of-principle experiments, we observed an impressive level of protection against stringent, high-dose heterologous SHIV challenges due to the synergy between a CD4mc and readily available recombinant vaccine candidates based on HIV-1$_{CH505}$ gp120. Anti-gp120 antibodies likely play an essential role in this protective immune response. BNM-III-170 sensitizes the SHIV-C5 challenge virus to neutralization and ADCC by these otherwise ineffectual antibodies, some of which have been shown to be elicited by Env-CD4 complexes that arise in Env vaccine recipients[25–27,35–38]. Vaccination with gp120, like most intact protein immunogens, elicits CD4[+] T-cell and antibody responses much more efficiently than CD8[+] cytotoxic T-cell responses[43,44]. The gp120-directed antibodies persist for at least six months after boosting, similar to the protective immune response (Supplementary Fig. 7). Even Env immunogens designed to elicit broadly neutralizing antibodies will likely generate other antibodies that neutralize virus or mediate ADCC only in the presence of CD4mc. Thus, CD4mc should increase the protective efficacy of any HIV-1 vaccine that elicits antibodies against Env and achieves less than 100% protection.

To contribute to HIV-1 prophylaxis, CD4mc need to interact with the virus and/or virus-infected cells at the time of exposure. The stoichiometry of CD4mc interaction with the Env trimer determines the impact on virus infectivity and sensitization of viruses and infected cells to antibodies (Supplementary Fig. 8)[45]. As some of the Env conformational changes induced by CD4mc are apparently irreversible[36], viruses or virus-infected cells that only transiently interact with CD4mc may still be vulnerable to

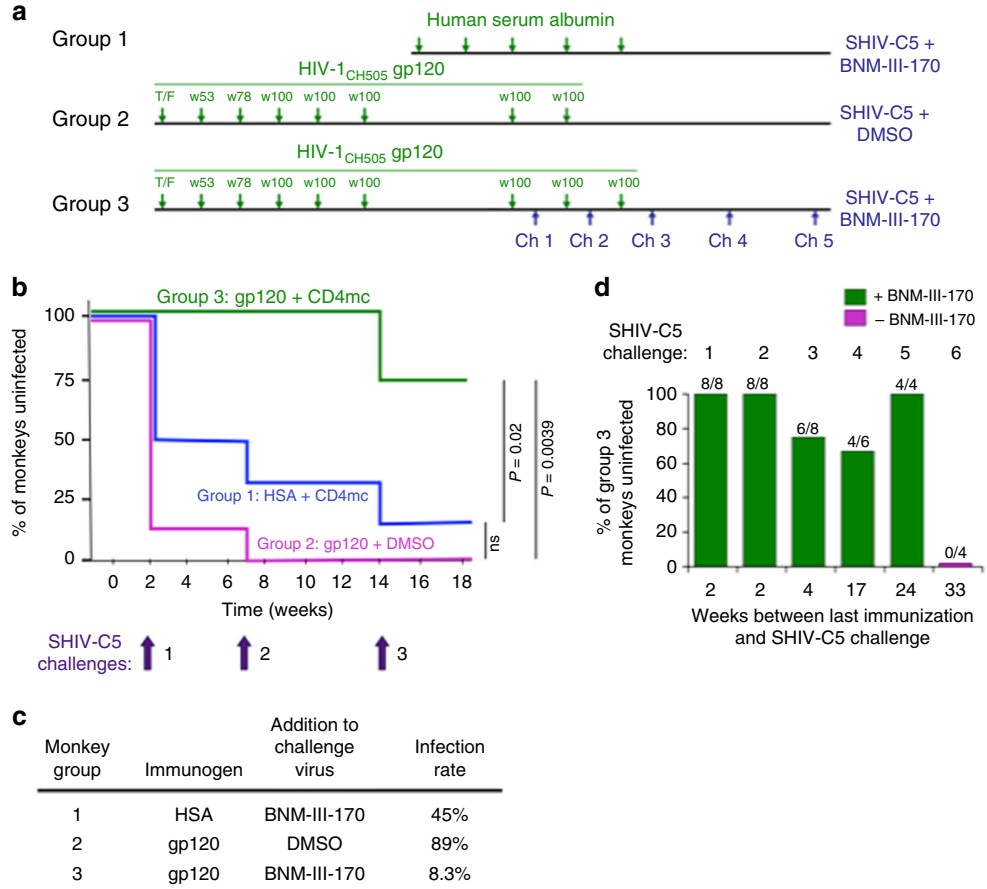

**Fig. 4** SHIV-C5 challenges and duration of protective immune responses. **a** Monkeys were immunized with either human serum albumin (Group 1) or gp120 glycoproteins corresponding to the Envs of the transmitted/founder virus and sequential virus isolates (at weeks 53, 78 and 100) from an HIV-1$_{CH505}$-infected individual (Groups 2 and 3) (green). Immunized monkeys were challenged with SHIV-C5 mixed with BNM-III-170 (Groups 1 and 3) or DMSO (Group 2) (blue). To accommodate the complete study in this schematic diagram, the time intervals on the horizontal axis are approximated. **b** Kaplan–Meier curves show the percentage of the monkeys in each group that remained uninfected after three high-dose intrarectal challenges with the heterologous Tier-2/3 transmitted/founder SHIV-C5[39]. The monkeys were boosted with human serum albumin (HSA) (Group 1) or HIV-1$_{CH505}$ gp120 (Groups 2 and 3) either two weeks (Challenges 1 and 2) or four weeks (Challenge 3) before the SHIV-C5 challenge. Infection of the Group 2 monkeys occurred at a rate expected for naive monkeys challenged with the same dose (3.5 animal infectious dose (AID$_{50}$) units) of SHIV-C5[39, 42]. The indicated P values were obtained using the log rank test (ns, not significant). The rate of infection of the Group 3 monkeys was significantly lower than that expected if gp120 immunization and BNM-III-170 treatment were simply additive ($P = 0.0186$, log rank test). **c** The results of the three SHIV-C5 challenges shown in (**b**) were used to calculate the infection rate, which is the number of infections/number of exposures. Compared with unvaccinated historical controls[39], the Jewell bias-corrected relative risk of SHIV-C5 acquisition in the Group 2 monkeys is 0.95 ± 0.24 (95% confidence interval). **d** The percentage of Group 3 monkeys that remained uninfected after SHIV-C5 challenges at different times following the last gp120 immunization is shown. The green bars indicate SHIV-C5 challenges that were conducted in the presence of BNM-III-170 (300 μM) and the magenta bar indicates a SHIV-C5 challenge performed in the absence of BNM-III-170. Each of the values obtained with BNM-III-170 significantly differs from that obtained in the absence of the CD4mc ($P < 0.05$, one-tailed Fisher exact probability test). The values obtained for the SHIV-C5 challenges performed with BNM-III-170 (green bars) do not significantly differ from each other

antibody attack, even distal from a mucosal site of exposure. In the particular experimental protocol used in this study, ADCC would be limited to SHIV-C5-infected cells sensitized by BNM-III-170 from the challenge virus inoculum; sustained mucosal or systemic administration of the CD4mc could greatly augment the protective potential of ADCC. The ability of CD4mc to synergize with vaccine-elicited antibodies to establish multiple barriers to HIV-1 transmission (blockade of CD4 binding, direct Env inactivation, virus neutralization and ADCC of infected cells) makes these small molecules unique among candidates for pre-exposure prophylaxis. These barriers should afford even greater protection in realistic contexts where the virus challenge doses are much lower. Plasma levels of BNM-III-170 at which biological effects on Env conformation can be readily detected are achievable in rhesus monkeys without evidence of toxicity (Evans, personal communication). Parallel development of the CD4mc, modalities

for sustained CD4mc delivery, and vaccines may allow the practical application of our results.

## Methods

**Study animals.** Adult Indian-origin rhesus macaques (Macaca mulatta) were used in this study. A PCR-based assay was used to determine the Mamu-A*01 status of the monkeys. Monkeys were housed at Bioqual, Inc., Rockville, MD. Monkeys were maintained in accordance with the Guide for the Care and Use of Laboratory Animals. All animal protocols were approved by the Institutional Animal Care and Use Committee.

**Synthesis and characterization of the CD4mc BNM-III-170.** BNM-III-170 [N1-(4-chloro-3-fluorophenyl)-N2-((1R,2R)−2-(guanidinomethyl)−5-((methylamino)methyl)−2,3-dihydro-1H-inden-1-yl) oxalamide] was synthesized as the bis-trifluoroacetate salt from commercially available 5-bromo-1-indanone following the procedures in ref. [31] (Fig. 1). Spectroscopic and physical data were in agreement with those in the cited work.

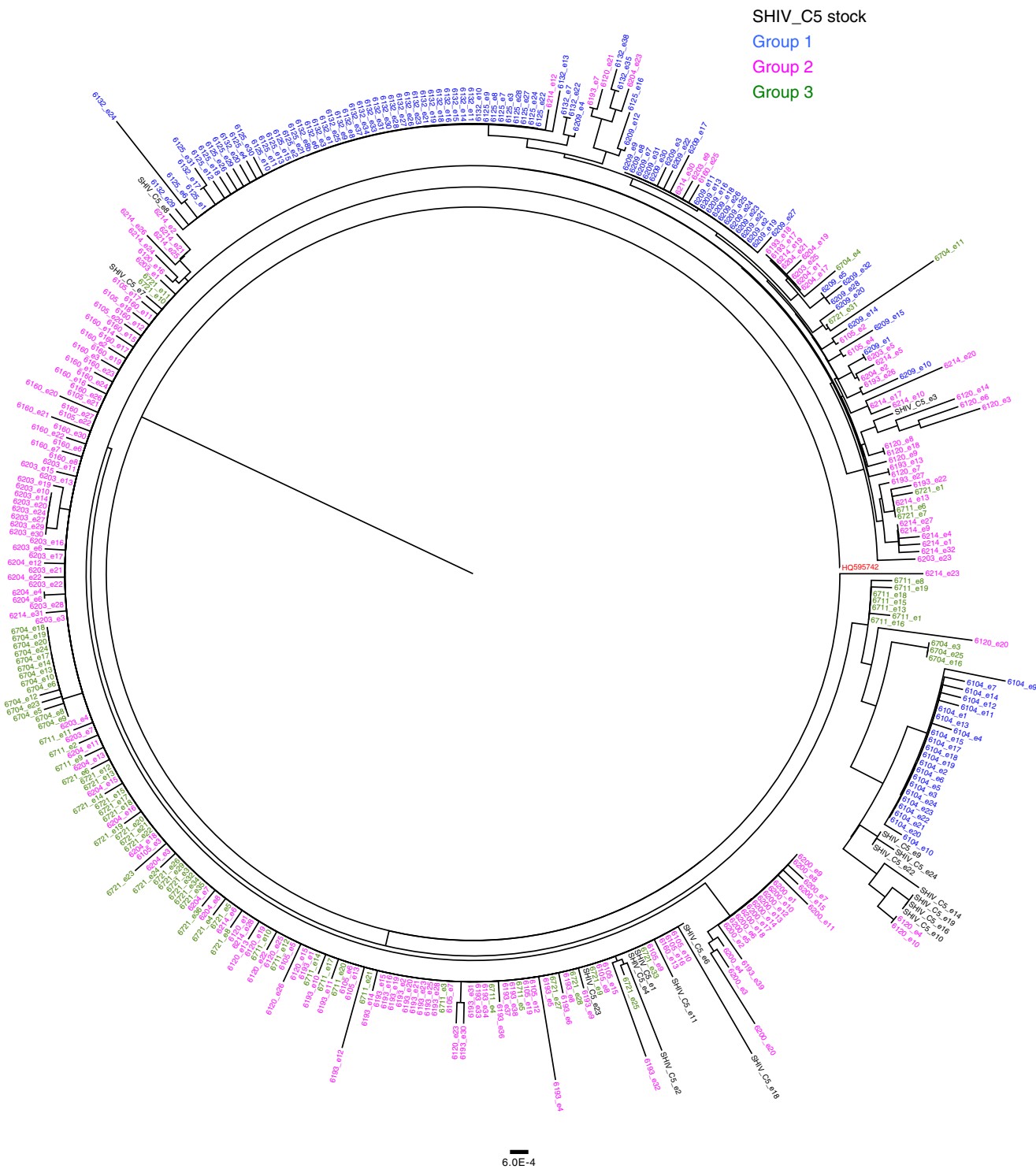

**Fig. 5** HIV-1 Env sequences of circulating SHIV-C5 in the infected monkeys. Plasma viral RNA was isolated from SHIV-C5-infected monkeys at the time of peak viremia and used for cDNA synthesis, single-genome amplification and env sequencing. The phylogenetic relationship of the env sequences is shown in a circular cladogram

**Cell lines**. 293T cells were obtained from the American Type Culture Collection. The cells were propagated in Dulbecco's Modified Eagle Medium with 10% fetal bovine serum, 8 mM L-glutamine and penicillin/streptomycin. Cf2Th-CD4/CCR5 cells were grown in Dulbecco's Modified Eagle Medium with 10% fetal bovine serum, 200 μg/ml hygromycin and 400 μg/ml G418 (Invitrogen). Cells were periodically tested to rule out mycoplasma contamination.

**Antibody**. The 17b antibody, which recognizes a CD4-induced epitope on HIV-1 gp120[46], was produced by transfection of 293T cells with plasmids expressing the

antibody heavy and light chains (a gift from Dr. James Robinson (Tulane University Medical Center)).

**Monkey immunizations**. An overview of the study is provided in Supplementary Fig. 1 and Supplementary Table 1.

Monkeys in Groups 2 and 3 were immunized as part of a previous study (CHAVI-ID NHP #109). In NHP #109, monkeys were sequentially immunized with the gp120 glycoproteins of viruses derived at multiple time points from an HIV-1$_{CH505}$-infected individual who generated a broad neutralizing antibody

response (ref. [47] and M. Anthony Moody and Barton Haynes, unpublished observations). The monkeys were immunized six times with gp120 in GLA-SE adjuvant (100 μg IM) 6-8 weeks apart. The CH505 gp120 glycoproteins in the six immunizations were derived from the transmitted/founder HIV-1$_{CH505}$ and from the viruses at weeks 53, 78, 100, 100 and 100. Five days after the first four immunizations, the monkeys received an intravenous infusion of basiliximab, M2850 anti-TacRh or the CH65 antibody. Basiliximab is a chimeric mouse-human monoclonal antibody that recognizes the alpha chain of the interleukin-2 receptor on T cells (Novartis)[48,49]. The M2850 anti-TacRh is a monoclonal antibody (rhesus macaque-modified) against the interleukin-2 receptor peptide (Tac)[50,51]. CH65 is a control human antibody (rhesus macaque-modified) against HA1 of influenza A virus[52]. No significant differences were observed in the titers of anti-gp120 antibodies elicited in the groups treated with the different monoclonal antibodies (Moody and Haynes, unpublished observations). Approximately 13 months elapsed between the last monoclonal antibody infusion and the first SHIV-C5 challenge.

Three groups of monkeys were used in the present study (CHAVI-ID NHP #124). Six naive monkeys in Group 1 were immunized twice with human serum albumin in GLA-SE adjuvant (100 μg IM). Sixteen monkeys from the CHAVI-ID NHP #109 study described above were divided into Groups 2 and 3 (8 monkeys/ group). All three groups of monkeys were matched with respect to Mamu A*01 alleles (Supplementary Table 1). Monkeys in Groups 2 and 3 were also matched as closely as possible with respect to the monoclonal antibody treatments received following the gp120 immunizations. Two weeks prior to SHIV Challenges 1 and 2 and four weeks prior to SHIV Challenge 3, the monkeys were boosted with 100 μg antigen IM (human serum albumin for Group 1 animals; CH505 wk 100 gp120 for Group 2 and 3 animals) in GLA-SE adjuvant. The monkeys were not boosted subsequently, prior to SHIV Challenges 4, 5, and 6.

**Recombinant viruses expressing luciferase.** To produce single-round HIV-1 expressing luciferase, 293T human embryonic kidney cells were cotransfected with plasmids expressing the pCMVΔP1Δenv HIV-1 Gag-Pol packaging construct, the HIV-1 envelope glycoproteins or the envelope glycoprotein of the control amphotropic murine leukemia virus (A-MLV), and the firefly luciferase-expressing vector at a DNA ratio of 1:1:3 μg using the Effectene transfection reagent (Qiagen) [35,36]. The plasmids expressing the HIV-1 envelope glycoproteins and Rev protein were based on pSVIIIenv[35] or pcDNA3.1 (Invitrogen Life Technologies, Carlsbad, CA). Cotransfection produced recombinant, luciferase-expressing viruses capable of a single round of infection. The virus-containing supernatants were harvested between 36 and 40 h after transfection and cleared of debris by low-speed centrifugation. Aliquots of the virus preparations were frozen at −80 °C until further use. The reverse transcriptase (RT) levels of all virus stocks were measured[53].

**Infection of single-round recombinant viruses.** Cf2Th-CD4/CCR5 target cells were seeded at a density of $6 \times 10^3$ cells/well in 96-well luminometer-compatible tissue culture plates (PerkinElmer) 24 h before infection. To test the direct antiviral activity of the CD4mc, on the day of infection, BNM-III-170 (0–100 μM) was incubated with recombinant viruses (10,000 RT units) at 37 °C for 30 min. For the sensitization assays, a constant concentration of BNM-III-170 (ranging from 15 to 50 μM) was incubated with virus for 30 min at 37 °C. Then the 17b antibody (over a range of 0–100 μg/ml) or monkey serum at different dilutions was added to the virus-compound mixture and incubated for an additional 30 min at 37 °C in a CO$_2$ incubator. The mixtures then were added to the target cells and incubated for 48 h at 37 °C. After this time, the medium was removed from each well and the cells were lysed by the addition of 30 μl passive lysis buffer (Promega) and three freeze-thaw cycles. An EG&G Berthold LB 96 V microplate luminometer was used to measure the luciferase activity in each well after the addition of 100 μl of luciferin buffer (15 mM MgSO$_4$, 15 mM KPO$_4$, pH 7.8, 1 mM ATP, and 1 mM dithiothreitol) and 50 μl of 1 mM 99% Firefly d-luciferin free acid (Prolume).

**ELISA.** The titer of anti-gp120 antibodies in the serum of immunized monkeys was measured by enzyme-linked immunosorbent assay (ELISA). For this purpose, 96-well ELISA plates (REACTI-BIND; Fisher Scientific) were coated with 100 μl of 2 μg/ml purified recombinant HIV-1$_{YU2}$ gp120 with a C-terminal His-6 tag, which was produced transiently by the transfection of 293F cells. ELISA plates coated with human serum albumin were used as a negative control. Coated plates were incubated at 4 °C overnight. Plates were washed three times with phosphate-buffered saline (PBS) containing 0.1% Tween 20 and blocked with 200 μl/well of 1% bovine serum albumin (BSA) in PBS and incubated at 37 °C for 1 h. Blocking buffer was aspirated, and serial dilutions of preimmune or immunized monkey sera were added in duplicate in a final volume of 100 μl/well and incubated at 37 °C for 1 h. Monkey sera were obtained on the day of SHIV-C5 Challenge 1. Plates were washed three times with PBS containing 0.1% Tween 20. Goat anti-human Fc gamma horseradish peroxidase (HRP) (Jackson ImmunoResearch Laboratories) was diluted 1:10,000 with blocking buffer, and 100 μl was added to each well and incubated at 37 °C for 1 h. Plates were washed three times with PBS containing 0.1% Tween 20. 3,3′,5,5′-Tetramethylbenzidine (TMB) single-solution substrate (Life Technologies) was added at a final volume of 100 μl/well and incubated at room temperature for 5 min to allow color to develop before reading at an optical density of 450 nm. The background of the ELISA was established by performing

the assay on the human serum albumin negative control without adding a primary antibody. As additional negative controls, the binding of uninfected monkey plasma and non-immune human sera to the gp120-coated plates was found to be equivalent to the background of the assay.

**Antibody recognition of infected cells and antibody-dependent cell cytotoxicity (ADCC).** Primary CD4$^+$ T lymphocytes were isolated by negative selection (EasySep human CD4+ T cell enrichment kit; STEMCELL) from resting peripheral blood mononuclear cells from three healthy HIV-1-negative individuals and were activated[37]. Cells were then infected with SHIV-C5 for 48 h before performing the antibody binding and ADCC assays.

Surface staining of infected cells was done with 1:250 dilutions of sera from monkeys obtained on the day of SHIV-C5 Challenge 1. The sera were tested in the presence of 50 μM BNM-III-170 or an equivalent volume of DMSO at 37 °C for 1 h. Goat anti-human antibody (AF-647) was used as the secondary antibody. The gates for the staining of SHIV-C5-infected cells by the monkey sera were established as shown in Supplementary Fig. 9. The mean fluorescence intensity (MFI) of AF-647 staining on Aqua Vivid (Invitrogen)-negative cells (living cells) was measured[8,37]. Mock-infected cells were used as a negative control, and the MFI of SHIV-C5-infected cells divided by the MFI of mock-infected cells is reported.

To evaluate ADCC, SHIV-C5-infected primary CD4$^+$ lymphocytes were incubated with autologous peripheral blood mononuclear cells (Effector:Target ratio of 10:1) from the same donor for 4–6 h at 37 °C in the presence of a 1:250 dilution of serum and 50 μM BNM-III-170 or equivalent volume of DMSO. Sera were obtained from the monkeys on the day of SHIV-C5 Challenge 1. Infected cells were identified by staining with an Alexa Fluor (AF)-488-conjugated antibody against SIV p27 CA. The percentage of ADCC killing was determined with the following formula: [(% of p27 $^+$ cells in Targets plus Effectors)−(% of p27 $^+$ cells in Targets plus Effectors plus serum)]/(% of p27 $^+$ cells in Targets) by gating infected (p27 $^+$) live (Aqua Vivid-negative) target cells[8,37]. In some experiments, 5 μg/ml of the Fab fragment of the A32 antibody was added to the ADCC assay.

**Challenge virus.** The simian-human immunodeficiency virus C5-1245045 (SHIV-C5) contains the Env from a South African Clade C transmitted/founder HIV-1[39]. The SHIV-C5 challenge virus stock was prepared by coculturing peripheral blood mononuclear cells (PBMC) from naive rhesus monkeys with CD8$^+$ T-cell-depleted PBMC and lymph node cells from infected monkeys as described[39]. The cell-free culture supernatants were harvested at days 14-21 and frozen in aliquots. The amount of p27 Gag, the viral RNA content and the infectivity of the stocks were measured[39]. The SHIV-C5 challenge stock contains $2.12 \times 10^8$ copies of viral RNA/ ml and 18,275 TCID$_{50}$/ml (measured on TZM-bl cells). The SHIV-C5 challenge stock contains approximately 7 intrarectal animal infectious doses (AID$_{50}$) per ml.

**Intrarectal challenges of monkeys.** All of the monkeys in this study were challenged with ~3.5 AID$_{50}$ SHIV-C5 applied intrarectally in an atraumatic fashion. At this dose, 91% of naive monkeys are expected to become infected after a single intrarectal inoculation[39,42].

For SHIV-C5 Challenges 1–5, 0.5 ml (~3.5 AID$_{50}$) of the SHIV-C5 stock was diluted to a total volume of 1 ml. For Group 1 and Group 3 monkeys, a volume of a 10 mM solution of the CD4mc BNM-III-170 in DMSO was added to achieve a final concentration of 300 μM BNM-III-170. For the Group 2 monkeys, the same volume of DMSO was added to the SHIV-C5 challenge stock. Within 30 min of its preparation, the virus-compound mixture was applied atraumatically to the monkey rectum. After 30 min in the prone position, the monkeys were allowed to recover from the anesthesia and returned to their quarters.

SHIV-C5 Challenge 6 was conducted in four Group 3 monkeys as described above, except that BNM-III-170 was not added to the SHIV-C5 stock prior to intrarectal challenge. Instead, the SHIV-C5 stock was incubated with an equivalent volume of DMSO prior to intrarectal inoculation.

**Plasma viral RNA measurements.** Measurements of viral RNA levels in the plasma of the SHIV-challenged monkeys were performed using a Qiagen QIA-symphony DSP Virus/Pathogen Midi Kit with the QIAsymphony SP platform and real-time PCR reaction[39]. The reactions were carried out on a StepOnePlus instrument (Applied Biosystems). This SIV viral load assay has been shown to be able to detect <250 viral RNA copies per ml. Monkeys that were detectably viremic on two occasions were considered infected.

**Statistical analysis.** Differences in the anti-gp120 antibody titers among the groups of immunized monkeys were evaluated with a Mann–Whitney U test. Differences in the serum neutralizing activity observed for viruses incubated with either BNM-III-170 or DMSO were evaluated using the "compare Growth Curves" function of the statmod software package for R (http://bioinf.wehi.edu.au/software/compareCurves). Differences in the ADCC-mediating activity among the groups of monkeys were evaluated by the one-way analysis of variance (ANOVA), with post-hoc analysis by Tukey's honest significant difference test[54].

As the potential effect of vaccination and BNM-III-170 sensitization of virus could not be predicted a priori, the number of monkeys in the study groups were maximized based on availability and resources. Monkeys were assigned to Groups

2 and 3 so that animals that received particular antibody treatments were distributed evenly between the groups. We also verified that the neutralization of BNM-III-170-sensitized HIV-1 by the sera of the Group 2 and Group 3 monkeys was comparable, prior to SHIV-C5 Challenge 1. The investigators were not blinded to the group allocation during the experiments. Following SHIV-C5 challenge, no monkeys were excluded from the analysis. Pairwise comparison of the percentage of monkeys that remained uninfected in Groups 1, 2, and 3 after the first three challenges, in which the hazard ratio is a constant, used the Kaplan–Meier method[41]. The log rank test[55] was used to test the null hypothesis that there is no difference between two groups of monkeys, and the test statistic was compared with a $X^2$ distribution with 1 degree of freedom[56].

A Fisher exact probability test was used to compare the percentage of Group 3 monkeys that remained uninfected after SHIV-C5 challenge at various times following gp120 immunization.

**Env sequence analysis**. The amino acid sequences of the HIV-1$_{CH505}$ gp120 used for monkey immunization and the Env of the SHIV-C5 challenge virus were compared by alignment in Clustal Omega[57]. The alignments were illustrated with ESPript 3[58].

Single-genome amplification (SGA) of cDNA from plasma viral RNA[59] was performed at the time of peak SHIV-C5 viremia. Briefly, a virus-containing pellet was obtained by ultracentrifugation of 1 ml of plasma from each animal at 24,000 × g for 3 h. The resulting pellet was suspended in 400 μl PBS and viral RNA was extracted by using the EZ1 Virus Mini Kit v2.0 (Qiagen, Valencia, CA) according to manufacturer's instructions. cDNA synthesis was performed using Superscript III Reverse Transcriptase (Invitrogen, Carlsbad, CA) and antisense primer 3R1 (5′-AGGKCTTTAAGCAAGCAAGCGTGGA-3′) located in the *nef* reading frame. The resulting cDNA suspension was diluted and PCR amplified with Platinum *Taq* DNA polymerase High Fidelity (Invitrogen) such that 25% of reactions were positive, thus maximizing the likelihood of single-genome amplification. A first round of PCR amplification was conducted using SHIVEnvR3-out 5′- CTA ATT CCT GGT CCT GAG GTG TAA TCC TG-3′ and SHIVEnvF4-out 5′- TCA TAT CTA TAA TCG TCA CGG AGA CAC TC −3′ as primers. A second round of PCR amplification was conducted using 2 μl of the first-round PCR product as template and SHIVEnvF2-in 5′- GTG TTG CTT TCA TTG CCA AGT TTG T −3′ and SHIVEnvR2-in 5′-TGG TAT GAT GCC TTC TTC CTT TTC T-3′ as primers. Amplification conditions for round 1 and round 2 PCR were 1 cycle of 94 °C for 2 min, 35 cycles of 94 °C for 15 s, 55 °C for 30 s, and 68 °C for 4 min, followed by 1 cycle of 68 °C for 10 min. Round 2 PCR amplicons were visualized by agarose gel electrophoresis, subsequently purified using a 0.6 × AMPure XP (Beckman Coulter, Indianapolis, IN) bead cleanup, and directly sequenced using an ABI3730xl genetic analyzer (Applied Biosystems). Full sequences were constructed from overlapping sequences from each amplicon in Sequencher program 5.4 (Gene Codes, Inc. Ann Arbor, MI). Sequence alignments and phylogenetic trees were constructed using clustalW and adjusted manually when necessary using SEAVIEW. Sites where there was a gap in any of the sequences, as well as areas of uncertain alignment, were excluded. Pairwise evolutionary distances were estimated by using the Kimura two-parameter model[60]. Phylogenetic trees were constructed by the neighbor-joining method[61].

The *env* genes of circulating viruses were amplified from the selected SGA-derived round-1 PCR amplicon for monkeys 6209 and 6125 in Group 3, using the *Taq* High Fidelity polymerase (Invitrogen Life Technologies, Carlsbad, CA). The *env* amplicons were then gel-purified and ligated into the pcDNA3.1 vector using the pcDNA3.1 Directional TOPO Expression Kit (Invitrogen Life Technologies, Carlsbad, CA)[62]. All *env* clones were confirmed by sequencing.

**Modeling CD4mc interactions with HIV-1 Env trimers**. The binding of CD4mc to the three protomers of the Env trimer was modeled using a binomial distribution[45]. The binomial distribution model assumes that each binding event of the CD4mc to the Env trimer occurs independently of the others. The binomial probability, $b$, of an Env trimer having $n$ protomers bound by the CD4mc is as follows:

$$b(n = 0) = \left( \frac{K_d}{[\text{CD4mc}] + K_d} \right)^3$$

$$b(n = 1) = \frac{3K_d^2[\text{CD4mc}]}{([\text{CD4mc}] + K_d)^3}$$

$$b(n = 2) = \frac{3K_d[\text{CD4mc}]^2}{([\text{CD4mc}] + K_d)^3}$$

$$b(n = 3) = \left( \frac{[\text{CD4mc}]}{[\text{CD4mc}] + K_d} \right)^3$$

where [CD4mc] is the concentration of the CD4mc and $K_d$ is the dissociation constant of the CD4mc for the Env trimer.

**Data availability**. Relevant data are available from the authors upon request. The DNA sequences of the *env* genes of the SHIV-C5 challenge stock and the circulating SHIVs in infected monkeys were deposited in GenBank (Accession numbers MH000783-MH001160).

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

## Acknowledgements

We thank Ms. Elizabeth Carpelan for manuscript preparation. This study was supported by the National Institutes of Health (AI124982, AI100645, and GM56550) and the late William F. McCarty-Cooper. N.M. was supported by amfAR grant 107431-45-RFNT, NIH AI90682 and a Ragon Institute Innovation Award. A.F. was supported by a Canada Research Chair and a CIHR foundation grant. J.R. and J.P. were supported by CIHR Fellowships and S.D. by an FRQS postdoctoral award.

## Author contributions

N.M., A.M.P., L.M., B.H., C.A.Z., B.P.C., L.S., T.B., M.A.M., S.D., J.P., J.R., A.F., B.F.H. S.S. and J.S. designed, performed, and analyzed experiments. B.M. and A.B.S. III synthesized and analyzed the CD4mc. N.M. and J.S. wrote the manuscript with input from the other authors.

## Additional information

**Competing interests:** The authors declare no competing interests.

