## [Peer Review File · Nature Communications]

Reviewers' comments:

Reviewer #1 (Remarks to the Author):

Medani et al. report on the vaccine-enhancing effects of a CD4 mimetic compound (CD4mc) mixed with the challenge virus in a nonhuman primate model. In line with their previous reports, they demonstrate that the CD4mc synergizes in vitro with antibodies elicited by immunization with monomeric HIV-1 gp120 and that gp120-immunized monkeys are protected from multiple high-dose intrarectal challenges with heterologous SHIV when the virus inoculum is pre-mixed with the CD4mc. The study was well conducted and written, but its significance remains uncertain.

Specific points:

1. Although the idea that a CD4mc can increase the protective efficacy of any HIV-1 Env-based vaccine is provocative, it is difficult to envisage a realistic applicability of this approach to prevent HIV-1 transmission. Administration of a CD4mc could conceivably be considered as post-exposure prophylaxis in people who may have received a prior Env-containing vaccine, which appears to be a pretty remote concurrence. Also, to reliably test this potential approach, the CD4mc should have been injected systemically prior to or immediately after the virus challenge, because a mucosal application would be ineffective unless performed at the time of sexual intercourse.
2. Although the mucosal application of a CD4mc in a fashion similar to topical microbicides could also be theoretically considered, it is equally unlikely to have practical applicability.
3. A more interesting approach would be to vaccinate with a CD4mc-Env complex, analogous to previous work (DeVico et al., PNAS 2007), possibly using VLPs or soluble Env trimers instead of gp120, which is notoriously ineffective in inducing protective immunity.
4. Besides the concerns regarding the applicability of this concept, the authors have not made any effort to elucidate the mechanism of protection in animals challenged with CD4mc-complexed virus. They state that the inherent antiviral effect of the CD4mc could contribute to protection, as seen in Group 1 animals, which further complicated the interpretation of the results. Furthermore, they did not investigate if the virus was completely neutralized at the port of entry, or rather blocked in the early stages of local dissemination by ADCC or other mechanisms. An extensive analysis of the early mucosal and local lymph node SHIV colonization, as well as its CD4mc-complexed status, would increase the significance of this study.

Reviewer #2 (Remarks to the Author):

The manuscript by Madini et. al., entitled “A CD4-mimetic compound enhances vaccine efficacy against stringent immunodeficiency virus challenge” is a relatively straightforward and quite interesting study that examines the protective capacity of a small molecular compound that induces gp120 conformational changes similar to CD4 (CD4mc), BNM-III-170, in combination with gp120 vaccine-elicited antibodies. The study examines BNM-II-170 as a proof-of-principle co-protective reagent, in the presence of commonly gp120-elicited CD4i antibodies following vaccination, and subsequent heterologous clade C SHIV challenge of non-human primates (NHPs).

The authors show quite convincingly that as stated on page 7, the “results suggest that the CD4mc...synergized with the gp120-elicited antibodies to protect monkeys from high-dose...challenges...” The statistical analysis appears to be well done.

Although this study is of interest and relevance to the HIV Env vaccine field, there are some limits to the logical and practical aspects to this pre-clinical study that need to be addressed.

a. If ADCC antibodies are protective in regards to HIV infection (naturally or vaccine-elicited), then why does one need the drug sensitization to reveal better this capacity? (If HIV escapes this activity by down regulation of gp120-CD4 complexes, then how is protective? How is this different from escape from V3 antibodies, for example).

To this reader, the putative protective effects of ADCC and in particular, anti-gp120 non-neutralizing antibodies, seem to be “modelled” as more likely to impact on viral load following infection and not so much relative to the acquisition of infection that is the primary endpoint of the current study.

b. The experimental design is a bit different than anticipated by this reviewer as the sensitizing drug needs to be pre-mixed with the challenge stock to mediate the protective effect reported in regards to acquisition of viral (SHIV) infection.

1. Fig 1. A schematic of the overall study design and the components in each of the different Groups would aid in clarity for a more general readership.

2. Of course, in the real world, the diverse challenge virus (circulating HIV-1 strains) won't be pre-incubated and co-delivered with the CD4mc drug. One assumes that the experiment was performed in this manner since there is still a “long way to go” for in vivo delivery of the CD4-like-inducing compound as an oral prophylactic? Otherwise, couldn't the NHPs have been treated with drug orally (or IV) immediately prior to challenge? The authors do briefly mention that for this

approach to be applied in the real world, a constant presences/level of the drug, following vaccination with some form of Env to elicit CD4i antibodies, would be needed, presumably in a slow release prophylactic modality?

3. Minor – The NHPs in Groups 2 and 3 for this study were immunized 6 times with CH505 gp120 in adjuvant, which is a considerable number of immunizations, and treated with antibodies for a previous experiment. Although this previous regimen is unlikely to affect the outcome reported here, it should be noted in the Main text.

c. Page 5-6, the antibodies that can mediate ADCC in vitro are dependent upon the presence of BNM-III-170. Presumably since they are cross-reactive between clade B and C these are 17b-like antibodies and not V3-directed? If so, the anti-sera could be tested in the Decker/Shaw HIV-2/CD4 cross-neutralization assay that can detect the presence of these antibodies. Could the protective capacity in vivo be due to this type of neutralizing activity rather than ADCC?

d. The animals were boosted with gp120 two weeks prior to the first challenge, which is quite soon after immunization. Apparently, the likely increase in CD4+T helper cells following the boost did not interfere with the observed protection, which is an interesting outcome.

1. Were gp120 binding (or neutralizing) titers measured at later time points following this boost prior to the later challenges? If there are serum or plasma samples available, this should be done. It is of interest, if this is due to the presence of circulating antibody, is this durability due to the multiple immunizations, the adjuvant? Antibody from memory B cells is not likely to be involved in the observed protection from mucosal challenge at the later time points?

Overall, this is a very interesting proof-of-principle study, but not without its limitations for real world applicability. This reviewer is an advocate of this line of investigation, however one could ask is this superior in some manner to PREP with the typical anti-virals? The authors do address the theoretical advantages of the CD4mc at the end of the paper, but this assumes in vivo bioavailability, lack of toxicity, etc.

Reviewer #3 (Remarks to the Author):

Summary

This manuscript evaluates the immunogenicity and efficacy of monomeric HIV-1 envelope (Env) immunization used in conjunction with a potential pre-exposure prophylactic compound that mimics CD4 binding to HIV-1 envelope. The central concept underlying the study is that CD4 mimetic compounds (CD4mc) expose Env epitopes that render the virus more susceptible to antibodies elicited by vaccination but that are not typically accessible due to steric hindrance upon CD4 binding. The hypothesis tested here is that CD4mc pre-treatment of virions containing HIV-1 Env augments the antiviral activity of Env-specific humoral immune responses elicited by vaccination in nonhuman primates. Sera from rhesus macaques vaccinated with gp120 is assessed in vitro for neutralization, binding, and ADCC activity on virus with and without CD4mc pretreatment. Vaccine efficacy is assessed by repeat high-dose challenge with SHIV-C5, an SIV chimeric virus expressing a subtype C HIV-1 transmitted/founder env, again with and without CD4mc pretreatment of the virus. While the SHIV challenge data presented appear promising, more rigorous statistical analysis of the challenge results is warranted. Immunologic characterization of the vaccinated animals is limited and does not substantiate major claims regarding the mechanism of protection. Together, these results represent an important first step in evaluating the ability of CD4 mimetic compounds to limit HIV-1 transmission; however, interpretation of the data is limited by the lack of appropriate control animals, immunogenicity measurements, and immune correlates analysis, along with several additional important technical issues noted below.

Major Issues: Key experiments required for acceptance

1) A major conclusion drawn by the authors from the results is that vaccine-elicited antibodies were responsible for reducing the rate of infection in vaccinated animals challenged with CD4mc pre-treated SHIV-C5 virus (e.g. page 7, line 173). At present, this interpretation is unsubstantiated and represents a major weakness of the manuscript. To attribute the vaccine-elicited antibodies to the reduced acquisition rate, immune correlates of risk analysis must be performed. If Env-specific antibodies were responsible, then the titers at the time of each challenge (1-5) would be expected to predict the likelihood of infection among animals in Group 3. If they do not correlate, then other alternate mechanisms of protection should be considered and presented.

2) The authors state that “protective antibody responses persist for at least six months after immunization,” (p. 3, line 80, p. 8, line 182), and yet no measurement of antibody responses beyond week 2 (time of challenge #1) is presented (Figures 1-2, Supp. Figure 1). It is conceivable that other adaptive immune responses contributed to the protection. Moreover, the slow rate of infection among the unvaccinated control arm (Group 1) revealed thus there is inherent variability to SHIV-C5+BNM-III-170 infection acquisition for independent of vaccination, with one animal (out of 6) resisting all 3 initial challenges. Without binding antibody responses present at each challenge and immune correlates analysis discussed above, the evidence that antibody responses persisted and mediated protection is weak and indirect. Longitudinal humoral responses to each gp120 vaccination

and spanning the entire challenge phase should be presented for all animals as measured by ELISA at a minimum.

3) Analysis of the challenge outcome data presented in Figure 3 does not conform to statistical standards established for evaluating efficacy in repeat challenge protection studies (Hudgens et al., *Biometrics* 2009). If the Hudgens et al. analysis approach is not employed, the authors should provide justification for why it does not apply to this study and rationale for use of the model employed. Vaccine efficacy should be calculated for each arm, using the appropriate pairwise comparator arm, and reported alongside the corresponding p-value. In addition, the authors comment that the “rate” of infection was lower in a given group compared to another group (e.g. p. 7, line 161), and yet no values for the rate of infection are presented. Infection rate, defined as the number of infections divided by the total number of exposures, should be calculated for each group. This reviewer finds it helpful to have this information presented in table form alongside the Kaplan-Meier curves in Fig 3A.

4) The lack of a control group of animals that were unvaccinated and challenged with untreated SHIV-C5 virus is a major weakness to the experimental design. While the primary comparison between vaccinated animals with and without CD4mc virus pretreatment does not rely upon unvaccinated controls, the efficacy of vaccination alone cannot be interpreted without this control arm. One possible, albeit unlikely, scenario that cannot be ruled out is that vaccination alone (Group 2) enhanced the infection rate. Reference to historical controls (Ref#37, Asmal et al.) is unsatisfying because historic controls are not a reliable substitute for contemporaneous challenges. In addition, the group size for the historic controls was very small (N=2). The authors should disclose this information within the Results, acknowledge this limitation of the study design, and include possible alternative interpretations of the efficacy data. The suggested table discussed above in point #3 should include the historical unvaccinated controls, noting their historic origin.

4) What is the safety profile of CD4mc when administered in vivo at an effective concentration? Does it bind natural CD4 ligands? Does it impact adaptive immunity or alter development of CD4 T-cell responses to vaccination? infection? The clinical relevance of this proof-of-principle study would be augmented by a summary of pre-clinical data demonstrating safe administration of effective concentrations of the compound to animals. Without demonstrating safety in animals or humans, the impact of this work and relevance for pre-exposure prophylaxis is reduced considerably.

5) The potential contribution of the passively transferred monoclonal antibodies to the in vitro immunogenicity assay results and challenge efficacy results should be discussed. This is particularly relevant given evidence that monoclonal antibodies have the capability to augment antigen presentation via formation of antigen-immune complexes presented on dendritic cells and enhance CD8 responses, resulting in HIV-1 virologic control and protection against influenza infection (Nishimura et al., *Nature* 2017; Leon et al., *J Ex Med* 2014). Concentration of the infused monoclonal

antibodies in each animal at the time of the first challenge should be presented or the authors should provide justification for why this data is not presented. HIV Env-specific CD8 T cell responses prior to immunization and at the time of challenge #1 should be reported.

Minor Issues:

1) The abstract focuses excessively on background information. Only the last 3 sentences describe the present study. More emphasis should be placed on the present study.

2) The concentration of BNM-III-170 used to treat the SHIV-C5 challenge virus differs from that used for the in vitro assays in Figures 1-2. Specifically, a much higher dose (300 μ M) was used for the challenge experiments while 20 and 50 μ M was used for the neutralization and ADCC assays, respectively. Rationale for the varying concentrations using throughout the study should be provided.

3) Figure 3B is confusing and largely redundant with Figure 3A. A more conventional and straightforward presentation of the challenge #4-6 data would be to present it as a continuation of the Kaplan-Meier curves presented in 3A, inclusive of the group 1 animal that became infected following challenge #4 mentioned in the text but not depicted in Figure 3.

4) Results from challenge #4 are presented on p. 7, line 165, without any prior mention that animals were challenged beyond 3 rounds. This result would be better placed in the paragraph beginning on p. 7, line 177 ("To examine the durability..."), following introduction of the fourth and fifth challenges.

5) The authors state that viremia was lower in the two Group 3 monkeys infected after the fourth SHIV challenge (p. 8, line 186). However, there is no data presented from the fourth SHIV challenge. Figure 4 only shows viremia following the first 3 challenges. The authors should present this data, specify what viremia measurement they compared (e.g. peak, set-point, area under curve), and provide statistics demonstrating a significant difference.

6) On a related note, the viral load data presented in Figure 4 has limited relevance and is largely not discussed in the manuscript. The viral load data is valuable and should be included in the publication, however it seems more appropriate for inclusion as supplemental data, barring addition of any

statistically significant differences in viral load that are not currently presented. In addition, a more meaningful presentation of the viral load data would include some minor modifications. First, displaying days post-infection on the x-axis would synchronize the curves relative to infection onset (time to infection is already presented in Figure 3). Animals that became infected by different challenges could be indicated by formatting symbols or in the figure legend. Synchronizing the data facilitates intra- and inter-group comparison of viral load values throughout all stages of infection. Second, the Group 3 graph should include viral load curves for all 8 animals following infection. Third, a summary line depicting the average viral load at each week post-infection by group with error bars would be helpful in capturing the overall viremia trends.

7) Description of the CH505 gp120 glycoprotein used for the six immunizations (page 20, line 437) is confusing. Do the time points listed (T/F, week 53, 78, 100, 100, and 100) correspond to each of the six immunizations, respectively? Please specify.

8) The study design and history of animal interventions is rather complicated. A schematic illustrating the timing of mAb infusion, vaccinations, and all challenges by group would be very helpful and should be presented as a figure panel.

9) In Supplementary Table 2, the title refers to Env from clades A, B, C and D HIV-1. However, the three HIV-1 Env depicted along the top of the table all appear to be of clade B or C origin. Reference to clades A and D should be removed from the title or data from additional Env from these clades should be added to the table.

Reviewer 1

Madani et al. report on the vaccine-enhancing effects of a CD4 mimetic compound (CD4mc) mixed with the challenge virus in a nonhuman primate model. In line with their previous reports, they demonstrate that the CD4mc synergizes in vitro with antibodies elicited by immunization with monomeric HIV-1 gp120 and that gp120-immunized monkeys are protected from multiple high-dose intrarectal challenges with heterologous SHIV when the virus inoculum is pre-mixed with the CD4mc. The study was well conducted and written, but its significance remains uncertain.

Specific points:

1. Although the idea that a CD4mc can increase the protective efficacy of any HIV-1 Env-based vaccine is provocative, it is difficult to envisage a realistic applicability of this approach to prevent HIV-1 transmission. Administration of a CD4mc could conceivably be considered as post-exposure prophylaxis in people who may have received a prior Env-containing vaccine, which appears to be a pretty remote concurrence. Also, to reliably test this potential approach, the CD4mc should have been injected systemically prior to or immediately after the virus challenge, because a mucosal application would be ineffective unless performed at the time of sexual intercourse.

Response: Our study was designed to test the concept that a small-molecule CD4-mimetic compound (CD4mc) could increase the protective efficacy of an HIV-1 Env vaccine. The reviewer asks whether this could be applied. If an HIV-1 Env vaccine is less than 100% protective (a very likely scenario), a small-molecule that increases vaccine efficacy would be highly desirable. The CD4-mimetic compound would need to be present, either locally or systemically, at or near the time of virus exposure. Given that CD4mc are small molecules, this could be achieved orally or by sustained release (vaginal rings, implants, depot). All of these delivery approaches are currently under development for pre-exposure prophylaxis of HIV-1 transmission.

2. Although the mucosal application of a CD4mc in a fashion similar to topical microbicides could also be theoretically considered, it is equally unlikely to have practical applicability.

Response: Methods for sustained release of antiviral agents in sexual mucosa are under development. In the absence of a safe and effective vaccine, these modalities for pre-exposure prophylaxis will become increasingly attractive.

3. A more interesting approach would be to vaccinate with a CD4mc-Env complex, analogous to previous work (DeVico et al., PNAS 2007), possibly using VLPs or soluble

Env trimers instead of gp120, which is notoriously ineffective in inducing protective immunity.

Response: The reviewer suggests using the CD4mc-Env complexes as vaccine immunogens. This would be expected to elicit even higher titers of antibodies against CD4-induced conformations of Env. We have previously shown that such antibodies are efficiently elicited in non-human primates and in humans by HIV-1 Envs that can bind host CD4 (Madani *et al.*, *J. Virol.* **90**:5031-5046 (2016)). Unfortunately, such antibodies do not prevent infection (unless the virus is first sensitized by the CD4mc, as shown for the first time in this study). For example, in the DeVico *et al.* study cited by the reviewer, 8/8 monkeys immunized with Env-CD4 complexes became infected after SHIV challenge, although some of the vaccinated monkeys exhibited lower viral loads than control animals.

4. Besides the concerns regarding the applicability of this concept, the authors have not made any effort to elucidate the mechanism of protection in animals challenged with CD4mc-complexed virus. They state that the inherent antiviral effect of the CD4mc could contribute to protection, as seen in Group 1 animals, which further complicated the interpretation of the results. Furthermore, they did not investigate if the virus was completely neutralized at the port of entry, or rather blocked in the early stages of local dissemination by ADCC or other mechanisms. An extensive analysis of the early mucosal and local lymph node SHIV colonization, as well as its CD4mc-complexed status, would increase the significance of this study.

Response: There was no statistically significant difference between the outcomes of the Group 1 and Group 2 monkeys. While the inherent antiviral effect of the CD4mc might explain the non-significant difference observed, this in no way complicates the interpretation of the major result reported by this manuscript, i.e., the Group 3 monkeys that received both the gp120 vaccine and the CD4-mimetic compound were protected from the SHIV challenge significantly better than either the Group 1 or Group 2 monkeys.

The reviewer suggested that an extensive analysis of the early mucosal and lymph node colonization by SHIV would be helpful in determining the stage at which the protection is mediated. Although this is an interesting topic, sampling of rectal mucosa or gut-associated lymphoid tissue was precluded by the study design. Any damage to these tissues might have altered the susceptibility of the monkeys to the SHIV challenge, which was applied atraumatically to the rectal mucosa. Therefore, in order not to compromise the study, we avoided any sampling of tissue from the mucosal challenge site. Future studies could potentially be designed to address the fate of the virus and virus-infected cells in the monkeys protected from infection. We note,

however, that virus particles or infected cells are expected to be very rare in the protected, uninfected monkeys, potentially making it difficult to reach definitive conclusions about the protective mechanism.

Reviewer 2

The manuscript by Madani et. al., entitled "A CD4-mimetic compound enhances vaccine efficacy against stringent immunodeficiency virus challenge" is a relatively straightforward and quite interesting study that examines the protective capacity of a small molecular compound that induces gp120 conformational changes similar to CD4 (CD4mc), BNM-III-170, in combination with gp120 vaccine-elicited antibodies. The study examines BNM-II-170 as a proof-of-principle co-protective reagent, in the presence of commonly gp120-elicited CD4i antibodies following vaccination, and subsequent heterologous clade C SHIV challenge of non-human primates (NHPs).

The authors show quite convincingly that as stated on page 7, the "results suggest that the CD4mc...synergized with the gp120-elicited antibodies to protect monkeys from high-dose...challenges..." The statistical analysis appears to be well done.

Although this study is of interest and relevance to the HIV Env vaccine field, there are some limits to the logical and practical aspects to this pre-clinical study that need to be addressed.

a. If ADCC antibodies are protective in regards to HIV infection (naturally or vaccine-elicited), then why does one need the drug sensitization to reveal better this capacity? (If HIV escapes this activity by down regulation of gp120-CD4 complexes, then how is protective? How is this different from escape from V3 antibodies, for example).

To this reader, the putative protective effects of ADCC and in particular, anti-gp120 non-neutralizing antibodies, seem to be "modelled" as more likely to impact on viral load following infection and not so much relative to the acquisition of infection that is the primary endpoint of the current study.

Response: The major HIV-1 Env conformation recognized by ADCC-mediating antibodies is the CD4-bound conformation. In HIV-1-infected cells, two viral proteins, Vpu and Nef, specifically interact with the cytoplasmic tail of CD4 and remove CD4-Env complexes from the cell surface. Thus, HIV-1-infected cells are relatively resistant to ADCC. However, because HIV-1 Vpu and Nef cannot interact with the CD4mc, CD4mc-Env complexes are displayed on the infected cell surface and can be targeted by the high titers of potentially ADCC-mediating antibodies in vaccinated or infected individuals.

The reviewer asked about HIV-1 escape from V3 antibodies. In contrast to escape from ADCC, where Vpu and Nef activity make Env epitope variation unnecessary, HIV-1 typically escapes from neutralization by V3-directed antibodies by altering the epitope.

The reviewer is correct that non-neutralizing antibodies mediating ADCC could potentially decrease viral loads by recognizing and destroying infected cells. However, such antibodies hypothetically could contribute to protection against virus acquisition if cells infected early after exposure were eliminated before significant release of virus occurred. In our experimental system, SHIV-infected cells would have to be sensitized to ADCC-mediating antibodies by the CD4mc in the challenge virus inoculum. Host cells initially infected by the challenge virus would require several hours to express Env, during which time the CD4mc would diffuse and be diluted. Given its dependence on the CD4mc, ADCC would be expected to be relatively inefficient in this setting. Therefore, virus-neutralizing antibodies likely contribute to the protection observed in the Group 3 monkeys.

b. The experimental design is a bit different than anticipated by this reviewer as the sensitizing drug needs to be pre-mixed with the challenge stock to mediate the protective effect reported in regards to acquisition of viral (SHIV) infection.

1. Fig 1. A schematic of the overall study design and the components in each of the different Groups would aid in clarity for a more general readership.

Response: This is a good suggestion. In addition to Supplementary Table 1, we provide Supplementary Figure 1 showing the immunization received by each monkey in the study and summarizing the results of the SHIV-C5 challenges.

2. Of course, in the real world, the diverse challenge virus (circulating HIV-1 strains) won't be pre-incubated and co-delivered with the CD4mc drug. One assumes that the experiment was performed in this manner since there is still a "long way to go" for in vivo delivery of the CD4-like-inducing compound as an oral prophylactic? Otherwise, couldn't the NHPs have been treated with drug orally (or IV) immediately prior to challenge? The authors do briefly mention that for this approach to be applied in the real world, a constant presences/level of the drug, following vaccination with some form of Env to elicit CD4i antibodies, would be needed, presumably in a slow release prophylactic modality?

Response: In this proof-of-principle experiment, the virus was mixed with the CD4mc to ensure contact of the infecting virus with the CD4mc. As discussed above, for real world application, sustained release of the CD4mc, either mucosal or systemic, would be

employed in vaccinated individuals. Pharmacokinetic and toxicity studies of CD4mc in monkeys are ongoing, and will inform future attempts to deliver these compounds for prevention and/or treatment of HIV-1 infection.

3. Minor – The NHPs in Groups 2 and 3 for this study were immunized 6 times with CH505 gp120 in adjuvant, which is a considerable number of immunizations, and treated with antibodies for a previous experiment. Although this previous regimen is unlikely to affect the outcome reported here, it should be noted in the Main text.

Response: The complete immunization regimen is now summarized in Supplementary Figure 1 and Supplementary Table 1. Examination of Supplementary Figure 1 shows that the outcome of the SHIV-C5 challenges was unrelated to the antibody treatment that the monkeys received.

c. Page 5-6, the antibodies that can mediate ADCC in vitro are dependent upon the presence of BNM-III-170. Presumably since they are cross-reactive between clade B and C these are 17b-like antibodies and not V3-directed? If so, the anti-sera could be tested in the Decker/Shaw HIV-2/CD4 cross-neutralization assay that can detect the presence of these antibodies. Could the protective capacity in vivo be due to this type of neutralizing activity rather than ADCC?

Response: The results in Figure 2C indicate that a substantial fraction of the antibodies that mediate ADCC in the presence of BNM-III-170 recognize gp120 epitopes in or near Cluster A. Cluster A epitopes are highly conserved among HIV-1 strains from different phylogenetic clades.

For the reasons discussed above, we expect that, in our experimental system, ADCC may be relatively inefficient. By contrast, all of the challenge virus bound by the CD4mc is potentially susceptible to neutralization by the monkey serum antibodies. Thus, neutralizing antibodies likely contributed to the observed protection in the Group 3 monkeys. These antibodies recognize CD4-induced gp120 epitopes conserved across Clades B and C. The antibodies that mediate neutralization of CD4mc-sensitized HIV-1 and the gp120 epitopes recognized have been characterized in previous publications from our group (Madani *et al.*, *J. Virol.* **88**:6542-6555 (2014); Madani *et al.*, *J. Virol.* **90**:5031-5046 (2016)).

d. The animals were boosted with gp120 two weeks prior to the first challenge, which is quite soon after immunization. Apparently, the likely increase in CD4+T helper cells following the boost did not interfere with the observed protection, which is an interesting outcome.

Response: We agree.

1. Were gp120 binding (or neutralizing) titers measured at later time points following this boost prior to the later challenges? If there are serum or plasma samples available, this should be done. It is of interest, if this is due to the presence of circulating antibody, is this durability due to the multiple immunizations, the adjuvant? Antibody from memory B cells is not likely to be involved in the observed protection from mucosal challenge at the later time points?

Response: We have added gp120 binding data from the later time points to the revised manuscript. Antibodies reactive with gp120 are readily detected in the serum of vaccinated monkeys 31 weeks after the last boost.

Overall, this is a very interesting proof-of-principle study, but not without its limitations for real world applicability. This reviewer is an advocate of this line of investigation, however one could ask is this superior in some manner to PREP with the typical anti-virals? The authors do address the theoretical advantages of the CD4mc at the end of the paper, but this assumes in vivo bioavailability, lack of toxicity, etc.

Response: The use of CD4mc for HIV-1 prophylaxis is unique among PreP regimens in that the CD4mc synergize with vaccines, a major conclusion of this manuscript. We emphasize this point in the revised Discussion.

Reviewer 3

This manuscript evaluates the immunogenicity and efficacy of monomeric HIV-1 envelope (Env) immunization used in conjunction with a potential pre-exposure prophylactic compound that mimics CD4 binding to HIV-1 envelope. The central concept underlying the study is that CD4 mimetic compounds (CD4mc) expose Env epitopes that render the virus more susceptible to antibodies elicited by vaccination but that are not typically accessible due to steric hindrance upon CD4 binding. The hypothesis tested here is that CD4mc pre-treatment of virions containing HIV-1 Env augments the antiviral activity of Env-specific humoral immune responses elicited by vaccination in nonhuman primates. Sera from rhesus macaques vaccinated with gp120 is assessed in vitro for neutralization, binding, and ADCC activity on virus with and without CD4mc pretreatment. Vaccine efficacy is assessed by repeat high-dose challenge with SHIV-C5, an SIV chimeric virus expressing a subtype C HIV-1 transmitted/founder env, again with and without CD4mc pretreatment of the virus. While the SHIV challenge data presented appear promising, more rigorous statistical analysis of the challenge results is warranted. Immunologic characterization of the vaccinated animals is limited and does not substantiate major claims regarding the mechanism of

protection. Together, these results represent an important first step in evaluating the ability of CD4 mimetic compounds to limit HIV-1 transmission; however, interpretation of the data is limited by the lack of appropriate control animals, immunogenicity measurements, and immune correlates analysis, along with several additional important technical issues noted below.

Major Issues: Key experiments required for acceptance

1) A major conclusion drawn by the authors from the results is that vaccine-elicited antibodies were responsible for reducing the rate of infection in vaccinated animals challenged with CD4mc pre-treated SHIV-C5 virus (e.g. page 7, line 173). At present, this interpretation is unsubstantiated and represents a major weakness of the manuscript. To attribute the vaccine-elicited antibodies to the reduced acquisition rate, immune correlates of risk analysis must be performed. If Env-specific antibodies were responsible, then the titers at the time of each challenge (1-5) would be expected to predict the likelihood of infection among animals in Group 3. If they do not correlate, then other alternate mechanisms of protection should be considered and presented.

Response: Our emphasis on the role of gp120-elicited antibodies in the observed protection is justified by the following observations:

- 1) The statistically significant difference between the outcomes in the Group 2 and Group 3 monkeys indicates that exposure of the challenge virus to BNM-III-170 was essential for protection! The high rate of infection in the Group 2 monkeys challenged with SHIV-C5 in the absence of BNM-III-170 underscores the dependence of the protection on the CD4mc. BNM-III-170 binds specifically to HIV-1 gp120 and sensitizes the virus to antibody neutralization and virus-infected cells to ADCC.
- 2) The immunization of monkeys with HIV-1 gp120 in GLA-SE elicits serum antibodies that neutralize HIV-1 and support ADCC of HIV-1-infected cells only in the presence of BNM-III-170. Of note, the gp120-reactive antibodies persist for at least six months after boosting, similar to the protective immune response. Immunization with gp120 in GLA-SE does not elicit detectable CD8-positive, MHC Class I-restricted cytotoxic T-cell responses. This is consistent with a number of studies demonstrating that protein immunogens like gp120 elicit antibodies and CD4-positive T-cell responses, but not CD8-positive T-cell responses (Morrison et al., J. Exp. Med. 163:903-921 (1986); Takahashi et al., Nature 344:873-875 (1990)).
- 3) In general, vaccine-elicited antibodies prevent infection, whereas T-cell responses do not prevent infections but reduce, control and clear pathogens.

In our experiments, we observe prevention of infection, consistent with antibody-mediated protection.

In the revised manuscript, we more explicitly describe the rationale behind our emphasis on the role of anti-Env antibodies in protection. Although both neutralizing and ADCC-mediating antibodies could formally contribute to the observed protection, as noted above, ADCC may be relatively inefficient in our experimental system. Therefore, antibodies that specifically neutralize the BNM-III-170-sensitized SHIV-C5 challenge virus likely contribute to the observed protection. Our results also do not rule out other immune mechanisms that hypothetically might contribute to the observed protection and, in the revised manuscript, we have more carefully worded our conclusions to reflect this fact.

The reviewer's suggestion to seek correlates of protection would be applicable in a much larger study. In the Hudgen *et al.* paper cited by the reviewer, the authors determine that at least 20 monkeys in each arm of a vaccine study are needed to have sufficient power to detect an immune surrogate of protection.

2) The authors state that “protective antibody responses persist for at least six months after immunization,” (p. 3, line 80, p. 8, line 182), and yet no measurement of antibody responses beyond week 2 (time of challenge #1) is presented (Figures 1-2, Supp. Figure 1). It is conceivable that other adaptive immune responses contributed to the protection. Moreover, the slow rate of infection among the unvaccinated control arm (Group 1) revealed thus there is inherent variability to SHIV-C5+BNM-III-170 infection acquisition for independent of vaccination, with one animal (out of 6) resisting all 3 initial challenges. Without binding antibody responses present at each challenge and immune correlates analysis discussed above, the evidence that antibody responses persisted and mediated protection is weak and indirect. Longitudinal humoral responses to each gp120 vaccination and spanning the entire challenge phase should be presented for all animals as measured by ELISA at a minimum.

Response: We have added anti-gp120 antibody titers from later time points. The slower rate of virus acquisition by the Group 1 monkeys, although not significantly different from the Group 2 monkeys, could reflect the direct antiviral effects of the CD4mc.

3) Analysis of the challenge outcome data presented in Figure 3 does not conform to statistical standards established for evaluating efficacy in repeat challenge protection studies (Hudgens et al., Biometrics 2009). If the Hudgens et al. analysis approach is not employed, the authors should provide justification for why it does not apply to this study and rationale for use of the model employed. Vaccine efficacy should be calculated for each arm, using the appropriate pairwise comparator arm, and reported alongside the

corresponding p-value. In addition, the authors comment that the “rate” of infection was lower in a given group compared to another group (e.g. p. 7, line 161), and yet no values for the rate of infection are presented. Infection rate, defined as the number of infections divided by the total number of exposures, should be calculated for each group. This reviewer finds it helpful to have this information presented in table form alongside the Kaplan-Meier curves in Fig 3A.

Response: Hudgens and Gilbert use maximum likelihood analysis to evaluate repeated low-dose virus challenges, where risk is low and partial risk reduction is the norm. Maximum likelihood methods are well suited to repeated low-dose challenge models. This does not negate the validity of our statistical analysis, which is applied to high-dose virus challenges. We apply the log-rank test to obtain nonparametric Kaplan-Meier estimates of survival functions. Such nonparametric estimates make no assumptions about the data except that the risk of infection is the same at each challenge (see below).

Although an evaluation of the efficacy of the gp120 vaccine was not a goal of this study, we have provided an estimate of the gp120 vaccine efficacy relative to historical controls, as discussed in greater detail below. We also provide infection rates for each group of monkeys in the revised Figure 3b.

4) The lack of a control group of animals that were unvaccinated and challenged with untreated SHIV-C5 virus is a major weakness to the experimental design. While the primary comparison between vaccinated animals with and without CD4mc virus pretreatment does not rely upon unvaccinated controls, the efficacy of vaccination alone cannot be interpreted without this control arm. One possible, albeit unlikely, scenario that cannot be ruled out is that vaccination alone (Group 2) enhanced the infection rate. Reference to historical controls (Ref#37, Asmal et al.) is unsatisfying because historic controls are not a reliable substitute for contemporaneous challenges. In addition, the group size for the historic controls was very small (N=2). The authors should disclose this information within the Results, acknowledge this limitation of the study design, and include possible alternative interpretations of the efficacy data. The suggested table discussed above in point #3 should include the historical unvaccinated controls, noting their historic origin.

Reponse: The reviewer correctly points out that the protective efficacy of the gp120 vaccine alone (Group 2) cannot be calculated without a control group of monkeys that were not vaccinated and challenged. However, as mentioned above, evaluating the efficacy of the gp120 vaccine was not a goal of our study. As the reviewer notes, the primary comparison between vaccinated animals without CD4mc (Group 2) and vaccinated animals with CD4mc (Group 3) does not rely upon unvaccinated controls.

Nonetheless, we can use historical controls to obtain an estimate of the protective efficacy of the gp120 vaccine, which is obviously low. Historical controls are appropriate in this case because the identical challenge virus stock was applied to rectal mucosa under identical conditions. Also, we can take advantage of the data from unvaccinated control monkeys that received different doses of the challenge virus to obtain an estimate of the animal infectious doses (AID₅₀) in the SHIV-C5 challenge stock; in this case, the data from all 7 unvaccinated monkeys can be utilized (not just 2 monkeys). We analyzed the results of the first SHIV-C5 challenge of the Group 2 and Group 3 monkeys, in comparison with the results of the SHIV-C5 challenge stock titration. The Jewell bias-corrected estimate of vaccine efficacy for the gp120 vaccine alone (Group 2) is 0.047, whereas the efficacy of the gp120 vaccine plus the CD4mc (Group 3) is close to 1.00. We have added an estimate of the relative risk of SHIV-C5 acquisition in the gp120-immunized (Group 2) monkeys to the Figure 3 legend of the revised manuscript.

4) What is the safety profile of CD4mc when administered in vivo at an effective concentration? Does it bind natural CD4 ligands? Does it impact adaptive immunity or alter development of CD4 T-cell responses to vaccination? infection? The clinical relevance of this proof-of-principle study would be augmented by a summary of pre-clinical data demonstrating safe administration of effective concentrations of the compound to animals. Without demonstrating safety in animals or humans, the impact of this work and relevance for pre-exposure prophylaxis is reduced considerably.

Response: The reviewer correctly notes that the practical application of CD4mc depends upon safely achieving effective concentrations in the relevant body compartments. In tissue culture assays, BNM-III-170 is not cytotoxic nor does it interfere with the interaction of CD4 with ligands other than HIV-1 gp120. Pharmacokinetic and toxicity studies of BNM-III-170 in mice and monkeys are in progress and will be reported elsewhere.

5) The potential contribution of the passively transferred monoclonal antibodies to the in vitro immunogenicity assay results and challenge efficacy results should be discussed. This is particularly relevant given evidence that monoclonal antibodies have the capability to augment antigen presentation via formation of antigen-immune complexes presented on dendritic cells and enhance CD8 responses, resulting in HIV-1 virologic control and protection against influenza infection (Nishimura et al., Nature 2017; Leon et al., J Ex Med 2014). Concentration of the infused monoclonal antibodies in each animal at the time of the first challenge should be presented or the authors should provide justification for why this data is not presented. HIV Env-specific CD8 T cell responses prior to immunization and at the time of challenge #1 should be reported.

Response: The passively transferred antibodies against the interleukin-2 receptor or the influenza virus HA1 exerted no detectable effect on the quantity or quality of the elicited anti-gp120 antibodies. Nonetheless, the Group 2 and Group 3 monkeys were matched with respect to exposure to these monoclonal antibodies at the time of vaccination. Given the known half-life of these monoclonal antibodies, their concentration in the monkey sera at the time of first SHIV-C5 challenge (13 months after the last antibody infusion) is expected to be negligible. As shown in the new Supplementary Figure 1, the outcome of the SHIV-C5 challenges was unrelated to the antibody treatment that the monkeys received. As mentioned above, the immunization of monkeys with HIV-1 gp120 in GLA-SE adjuvant does not elicit detectable Env-specific CD8-positive T-cell responses.

Minor points:

1) The abstract focuses excessively on background information. Only the last 3 sentences describe the present study. More emphasis should be placed on the present study.

Response: We feel that it is important to provide the general reader with sufficient background information in the abstract to understand why we performed the study and what the results mean. The abstract allows the reader to appreciate how CD4-imitic compounds work and why they synergize with an Env vaccine-elicited immune response to achieve protection.

2) The concentration of BNM-III-170 used to treat the SHIV-C5 challenge virus differs from that used for the in vitro assays in Figures 1-2. Specifically, a much higher dose (300 μ M) was used for the challenge experiments while 20 and 50 μ M was used for the neutralization and ADCC assays, respectively. Rationale for the varying concentrations using throughout the study should be provided.

Response: A BNM-III-170 dose higher than that required to trigger HIV-1 Env conformational changes in tissue culture assays was used for incubation with the SHIV-C5 challenge virus. This concentration of BNM-III-170 was used to ensure that a high proportion of the SHIV-C5 Env spikes would have the compound bound.

3) Figure 3B is confusing and largely redundant with Figure 3A. A more conventional and straightforward presentation of the challenge #4-6 data would be to present it as a continuation of the Kaplan-Meier curves presented in 3A, inclusive of the group 1 animal

that became infected following challenge #4 mentioned in the text but not depicted in Figure 3.

Response: The reviewer feels that Figure 3B is redundant with Figure 3A and is therefore confusing. The reviewer suggests merging all of the challenge data into a single Kaplan-Meier graph. Although this seems like a reasonable way to present the data, there is an important reason not to do so. An assumption underlying Kaplan-Meier analysis is that the risk of infection in each group of monkeys does not change over the time period analyzed. Because the Group 2 and Group 3 monkeys were boosted prior to each of Challenges 1-3, this assumption is reasonable. This assumption is not reasonable for SHIV-C5 challenges 4 and 5, which occurred 17 and 24 weeks, respectively, after the previous boost. Therefore, these results need to be presented separately from the Kaplan-Meier plot of the initial three SHIV-C5 challenges. [As an aside, we tested the Kaplan-Meier statistics of all the data from Challenges 1-5 combined; with the caveat that the assumptions underlying the log-rank test were likely invalid, the conclusions were unchanged.]

4) Results from challenge #4 are presented on p. 7, line 165, without any prior mention that animals were challenged beyond 3 rounds. This result would be better placed in the paragraph beginning on p. 7, line 177 (“To examine the durability...”), following introduction of the fourth and fifth challenges.

Response: This is a good suggestion, and we have modified the text accordingly.

5) The authors state that viremia was lower in the two Group 3 monkeys infected after the fourth SHIV challenge (p. 8, line 186). However, there is no data presented from the fourth SHIV challenge. Figure 4 only shows viremia following the first 3 challenges. The authors should present this data, specify what viremia measurement they compared (e.g. peak, set-point, area under curve), and provide statistics demonstrating a significant difference.

Response: All of the group-averaged viremia data is now presented in Supplementary Figure 5. As the overall levels of viremia do not differ among the three groups of monkeys, we don't feel that it is necessary to mention the results of Challenge 4 specifically.

6) On a related note, the viral load data presented in Figure 4 has limited relevance and is largely not discussed in the manuscript. The viral load data is valuable and should be included in the publication, however it seems more appropriate for inclusion as supplemental data, barring addition of any statistically significant differences in viral load

that are not currently presented. In addition, a more meaningful presentation of the viral load data would include some minor modifications. First, displaying days post-infection on the x-axis would synchronize the curves relative to infection onset (time to infection is already presented in Figure 3). Animals that became infected by different challenges could be indicated by formatting symbols or in the figure legend. Synchronizing the data facilitates intra- and inter-group comparison of viral load values throughout all stages of infection. Second, the Group 3 graph should include viral load curves for all 8 animals following infection. Third, a summary line depicting the average viral load at each week post-infection by group with error bars would be helpful in capturing the overall viremia trends.

Response: These are good suggestions, which we have implemented in the revised manuscript. The viral load data is now presented in Supplementary Figures 4 and 5. Displaying all of the viremia data in a synchronized fashion would have more value if there were meaningful differences in the levels of viremia among the groups. However, such a presentation is very complicated and doesn't allow the reader to evaluate the viral loads for each individual monkey. Therefore, we have kept the viral loads for each challenge plotted separately in Supplementary Figure 4. We have, however, generated average viremia curves for each group, as suggested by the reviewer. These are presented in a synchronized fashion in the new Supplementary Figure 5.

7) Description of the CH505 gp120 glycoprotein used for the six immunizations (page 20, line 437) is confusing. Do the time points listed (T/F, week 53, 78, 100, 100, and 100) correspond the each of the six immunizations, respectively? Please specify.

Response: Yes, the gp120 glycoproteins used for each of the six immunizations were derived from HIV-1 variants present in the CH505-infected individual at the indicated time points. The newly added Supplementary Figure 1 should help to clarify this.

8) The study design and history of animal interventions is rather complicated. A schematic illustrating the timing of mAb infusion, vaccinations, and all challenges by group would be very helpful and should be presented as a figure panel.

Response: This is a good suggestion and we have included Supplementary Figure 1 in the revised manuscript summarizing the study design, history and outcomes.

9) In Supplementary Table 2, the title refers to Env from clades A, B, C and D HIV-1. However, the three HIV-1 Env depicted along the top of the table all appear to be of clade B or C origin. Reference to clades A and D should be removed from the title or data from additional Env from these clades should be added to the table.

Response: This has been corrected in the revised manuscript.

Finally, we have analyzed the sequence and phenotypes of the breakthrough SHIV-C5 viruses that infected the monkeys. These studies revealed some heterogeneity in the SHIV-C5 challenge stock and showed that the infecting viruses remained sensitive to BNM-III-170 and to the neutralizing antibodies in the serum of the challenged monkeys. Thus, infection appears to have occurred stochastically. These results provide some insights into the events surrounding the mucosal infections in these monkeys.

We thank the reviewers for their helpful suggestions. We trust that the revised manuscript is now suitable for publication in Nature Communications.

Sincerely,

Joseph Sodroski

Reviewers' comments:

Reviewer #1 (Remarks to the Author):

Reviewer 1

In their response, Madani et al. addressed all the issues raised in the initial review, but several questions remain on the overall significance of this study.

Specific points:

1. Regarding the applicability of this approach to prevention of HIV-1 transmission, the only plausible scenario appears to be a 'topical microbicide'-like application at the time of intercourse or through a vaginal ring or implant. However, the safety of mucosal delivery remains questionable, as is the possible induction of specific anti-CD4mc immune responses. On the other side, continuous systemic delivery of the CD4mc (by oral or parenteral routes) seems impractical and potentially unsafe.

2. See above.

3. There is no doubt that using the CD4mc-Env complexes as vaccine immunogens would elicit even higher titers of antibodies against CD4-induced conformations of Env, but this seems to be the very goal of the authors who are proposing the use of the CD4mc to increase the protective efficacy of HIV vaccines. Since the CD4mc unravels CD4-induced conformations of Env, such antibodies would be even more efficient in neutralizing CD4mc-complexed virus, even if ineffective against unliganded virus. If DeVico et al. had used a CD4mc pre-mixed with their virus challenge, their vaccine would have been highly protective.

4. Several questions remain regarding the mechanism of vaccine-induced protection in animals challenged with CD4mc-bound virus. First, the inherent antiviral effect of the CD4mc does complicate the data interpretation because at least part of the protective effect in Group 3 could be due to such antiviral effect, possibly acting in synergy with vaccine-induced antibodies or other mechanisms. For example, the authors show that vaccine-induced antibodies bind Env trimers more efficiently after incubation with the CD4mc, but any trimer that carries bound CD4mc is by definition blocked by the CD4mc. As stated in the first review, no efforts were made by the authors to precisely identify the mechanism of protection or at least to establish protection correlates. Third, although this reviewer acknowledges that early biopsies could damage mucosal tissues and alter the

susceptibility of the monkeys to SHIV challenge, biopsies could be performed after the 48 hours window for mucosal transmission or, even better, on a separate group of animals.

Reviewer 2

The authors have satisfactorily addressed many of the points raised in the first review. However, the logical and practical aspects of this vaccine-boosting concept require further investigation.

Specific points:

Point a. Authors' response: The major HIV-1 Env conformation recognized by ADCC-mediating antibodies is the CD4-bound conformation. In HIV-1-infected cells, two viral proteins, Vpu and Nef, specifically interact with the cytoplasmic tail of CD4 and remove CD4-Env complexes from the cell surface. Thus, HIV-1-infected cells are relatively resistant to ADCC (...) ADCC would be expected to be relatively inefficient in this setting. Therefore, virus-neutralizing antibodies likely contribute to the protection observed in the Group 3 monkeys.

This point is controversial. When CD4 T cells are infected in vitro, antibodies to cluster-A epitopes, like A32, are the first and most prominently reactive with the infected cell surface. In fact, it takes several days for Vpu and Nef to fully downregulate CD4. Thus, recently infected CD4 T cells should be excellent targets for ADCC in vivo, and it remains unclear why ADCC-mediating antibodies, if protective, should not work efficiently even in the absence of CD4mc. Regarding ADCC, the authors' statement that "ADCC would be expected to be relatively inefficient in this setting" seems to contradict the emphasis given to ADCC as a potential mechanism of protection in the manuscript. Conversely, the role of virus-neutralizing antibodies in protection is somewhat difficult to envisage here since binding to the CD4 mimetic would induce the trimer to assume an open conformation and make the virus non-infectious.

Point b. Authors' response: This is a good suggestion. In addition to Supplementary Table 1, we provide Supplementary Figure 1 showing the immunization received by each monkey in the study and summarizing the results of the SHIV-C5 challenges.

This point has been satisfactorily addressed.

Point c. Authors' response: In this proof-of-principle experiment, the virus was mixed with the CD4mc to ensure contact of the infecting virus with the CD4mc. As discussed above, for real world application, sustained release of the CD4mc, either mucosal or systemic, would be employed in vaccinated individuals. Pharmacokinetic and toxicity studies of CD4mc in monkeys are ongoing, and will inform future attempts to deliver these compounds for prevention and/or treatment of HIV-1 infection.

This remains a major hurdle in the practical development of this vaccine-boosting concept. Pharmacokinetic and toxicity/immunogenicity studies are mandatory to draw any conclusion on the potential applicability of this approach.

Minor points:

The authors have satisfactorily addressed all the minor points.

Reviewer #3 (Remarks to the Author):

The authors have adequately addressed many of this reviewer's specific comments from the original review, with some new or remaining outstanding minor points listed below. Overall, this work represents an important proof-of-principal animal protection study for a novel and interesting concept, that of sensitizing virus to vaccine-elicited antibodies using a small compound that simulates CD4 binding to HIV-1 Env. But the larger question pertaining to utility is difficult to overlook. The broader impact of this work in the field will likely be somewhat limited due to the practical challenge of applying this strategy as an HIV-1 prophylaxis, since it would require delivery of CD4mc to sites of HIV-1 exposure to treat and render virus susceptible to vaccine elicited antibodies.

Major limitations of the study related to study design also hamper its potential impact, namely: 1) the unclear effect of CD4mc alone on virus infectivity (group 1) confounds interpretation of the group 3 efficacy, 2) the vaccine regimen consisted of seven gp120 protein immunizations prior to the first challenge, and the gp120 sequence differed for each of the first four immunizations, a regimen difficult to envisage clinically, and 3) protein boosting shortly before each of the first 3 challenges. Their potential impact on the primary finding of macaque protection against SHIV challenge could be substantial, but cannot be discerned from the present study.

Minor points

- 1) The addition of gp120-specific binding antibody data in Supplemental Figures 2 and 7 adequately address the question of vaccine-elicited antibody durability. However, why do the ELISA values differ for the day of challenge #1 time point between Figures 2 and 7? Also, it would be helpful present a summary of the Figure 7 longitudinal responses in a single panel within this figure to illustrate the response decay kinetics in Group 3. The responses appear unusually stable with little to no decrease during the 6 months post-vaccination. The authors should comment on what may have contributed to this durability. The nine protein vaccinations?
- 2) It appears that Groups 2 and 3 were not entirely balanced with respect to the passively transferred antibodies administered during the first four gp120 vaccinations. For example, from Supp. Figure 1, M2850 was administered to 2 animals in G2 and 4 in G3.
- 3) Supplemental Figure 1 is an important addition to the manuscript, and attempts to clarify many details about the history and genetic background of each animal. The authors are commended for fully disclosing these details. However, this reviewer finds the table quite complicated and extremely difficult to follow. A simple timeline summarizing the entire protocol within the Main Figure 3 would improve transparency regarding the full vaccination and challenge schedule, including indication of the ongoing vaccinations during challenges 1-3. Burying the vaccination regimen and timing relative to the challenges in Supplemental is a disservice to readers.
- 4) The main text of the manuscript fails to mention that different gp120 immunogens were administered over the course of the seven pre-challenge vaccinations. This is misleading, as the current text suggests that a single gp120 immunogen was used, the CH505 transmitted founder Env, when in fact that was only used once for the first vaccination. It is an important technical detail that should be stated in the Main manuscript, and included in the vaccination timeline presented in Figure 3, as suggested above. It raises possible questions about whether multiple Env variants would be a requirement for this vaccination to be effective.
- 5) While a formal immune correlates analysis may not be feasible with a group size of 8, it is noteworthy that the gp120-specific titers in the Group 3 animals at the time of first challenge appear to show no association with time to infection. If true, the authors should acknowledge this observation in the Results and offer an explanation of why this is not incongruous with their proposed mechanism of protection in the Discussion.

Gavin Mason, Ph.D.
Associate Editor
Nature Communications

Dear Dr. Mason,

We were pleased to see that the reviewers of our revised manuscript "A CD4-mimetic compound enhances vaccine efficacy against stringent immunodeficiency virus challenge" (NCOMMS-17-19430A) felt that we satisfactorily addressed many of their comments. The reviewers raised some additional issues, to which we provide a point-by-point response as follows:

Reviewer 1

In their response, Madani et al. addressed all the issues raised in the initial review, but several questions remain on the overall significance of this study.

Specific points:

1. Regarding the applicability of this approach to prevention of HIV-1 transmission, the only plausible scenario appears to be a 'topical microbicide'-like application at the time of intercourse or through a vaginal ring or implant. However, the safety of mucosal delivery remains questionable, as is the possible induction of specific anti-CD4mc immune responses. On the other side, continuous systemic delivery of the CD4mc (by oral or parenteral routes) seems impractical and potentially unsafe.

Response: We agree that a vaginal ring or implant would be an attractive sustained delivery modality of CD4-mimetic compounds (CD4mc). At this early stage of development, we cannot rule out other delivery modes. The reviewer (and Reviewer 2) correctly noted that the safety of the CD4mc will have a major impact on the potential applicability of this approach. As we mentioned in our previous rebuttal letter, pharmacokinetic and toxicity studies of CD4mc in monkeys are ongoing. Here we provide a summary of the current results of these studies for the reviewers:

Pharmacokinetic/toxicity studies of BNM-III-170

A dose-escalation study of BNM-III-170 was performed in two rhesus monkeys. Monkeys were inoculated with a single subcutaneous injection of 3, 6, 12 or 24 mg/kg BNM-III-170 in 5% DMSO. Serum levels of BNM-III-170 were measured, and an assay was established to measure the ability of the BNM-III-170 in the monkey serum to induce conformational changes in Env expressed on the surface of HIV-1-infected cells. Blood chemistry and hematology were monitored, and electrocardiograms were recorded during administration of the compound. BNM-III-170 levels in the serum peaked within one hour of administration and reached 9.7 µg/ml at the 24 mg/kg dose. The serum half-life of BNM-III-170 was 3-6 hours, similar to that of several other antiretroviral drugs. The ability of the BNM-III-170 in the serum to induce Env conformational changes could be detected 24 hours after the 24 mg/kg dose and 20 hours after the 12 mg/kg dose. BNM-III-170 was well-tolerated at all doses tested to date. No clinical signs of toxicity were observed. Hematology tests and electrocardiograms were within normal limits. Serum chemistry values were also within normal limits, except for transient elevations of CPK, SGOT and LDH associated with the intramuscular administration of the ketamine anesthetic. Troponin I levels were monitored to rule out cardiac toxicity and remained low throughout the study. Thus, BNM-III-170 levels in the range at which biological effects on Env conformation are readily detected can be achieved in rhesus monkeys without evidence of toxicity.

These pharmacokinetic/toxicity studies will be extended and will be published at a future date. The apparent safety of systemically administered BNM-III-170 increases our confidence that local delivery via vaginal rings or implants will be feasible and well-tolerated.

The reviewer mentions the possible induction of specific anti-CD4mc immune responses. The formula weight of BNM-III-170 is less than 700 Daltons. Thus, although allergic or hapten-mediated immune responses are possible, BNM-III-170 is no more likely to induce these responses than are other small-molecule compounds. Future studies will evaluate whether or not this possibility represents a practical problem.

2. *See above.*

3. *There is no doubt that using the CD4mc-Env complexes as vaccine immunogens would elicit*

even higher titers of antibodies against CD4-induced conformations of Env, but this seems to be the very goal of the authors who are proposing the use of the CD4mc to increase the protective efficacy of HIV vaccines. Since the CD4mc unravels CD4-induced conformations of Env, such antibodies would be even more efficient in neutralizing CD4mc-complexed virus, even if ineffective against unliganded virus. If DeVico et al. had used a CD4mc pre-mixed with their virus challenge, their vaccine would have been highly protective.

Response: We agree.

4. Several questions remain regarding the mechanism of vaccine-induced protection in animals challenged with CD4mc-bound virus. First, the inherent antiviral effect of the CD4mc does complicate the data interpretation because at least part of the protective effect in Group 3 could be due to such antiviral effect, possibly acting in synergy with vaccine-induced antibodies or other mechanisms. For example, the authors show that vaccine-induced antibodies bind Env trimers more efficiently after incubation with the CD4mc, but any trimer that carries bound CD4mc is by definition blocked by the CD4mc. As stated in the first review, no efforts were made by the authors to precisely identify the mechanism of protection or at least to establish protection correlates. Third, although this reviewer acknowledges that early biopsies could damage mucosal tissues and alter the susceptibility of the monkeys to SHIV challenge, biopsies could be performed after the 48 hours window for mucosal transmission or, even better, on a separate group of animals.

Response: We agree that part of the protective effect in the Group-3 monkeys could be due to the direct antiviral effect of the CD4mc. However, this does not affect the validity of our conclusions. In general, the fact that prevention modalities A and B individually exhibit some efficacy does not preclude a demonstration that they are synergistic. The data show that the Group-3 monkeys (vaccine + CD4mc) were protected significantly better than either the Group-1 monkeys (CD4mc alone) or the Group-2 monkeys (vaccine alone) (Figure 3a). The level of protection against this stringent SHIV challenge in the Group-1 monkeys (CD4mc alone) did not significantly differ from that of the Group-2 monkeys (vaccine alone) ($P=0.32$), and neither the Group-1 monkeys nor the Group-2 monkeys were protected better than expected for a group of naïve monkeys. The degree of protection in the Group-3 monkeys was significantly better than that expected if the vaccine and the CD4mc were simply additive ($P=0.0186$). These results support the conclusion that, regardless of the possible direct antiviral effects of the CD4mc, the

protection observed in the Group-3 monkeys indicates synergy between the CD4mc and the gp120 vaccine.

Although the revised manuscript contained additional information that provided insights into the events associated with protection or breakthrough, we apparently did an inadequate job of relating this information to a mechanistic model. Both Reviewer 1 and Reviewer 2 expressed some difficulty in understanding the mechanism of the observed synergy between the CD4mc and the vaccine-induced antibodies. The comments of these reviewers helped us appreciate the basis of these difficulties. Both reviewers envisage an incorrect model in which "a trimer that carries bound CD4mc is by definition blocked by the CD4mc." If such a model were true, then the observed CD4mc sensitization of HIV-1 to neutralization by otherwise ineffectual antibodies could not occur. The data indicate otherwise, underscoring the inadequacy of a mechanistic model in which binding of a single CD4mc to the Env trimer renders the trimer non-functional.

In the revised manuscript, we interpret our results in terms of a more accurate mechanistic model that is consistent with all of the available data. This model is grounded in previously published work (Madani et al. *J. Virol.* 91: e10880-16 (2017)), and we refer the readers to this paper for details. The key feature of this model is that the consequences of the binding of CD4mc to the HIV-1 Env trimer depend upon stoichiometry, i.e., how many of the three potential CD4mc binding sites on each trimer are actually occupied. We have shown that HIV-1 Env trimers with a single CD4mc bound are infectious and potentially sensitized to antibody neutralization; by contrast, Env trimers with all three protomers bound by CD4mc are inactivated. Env trimers with two CD4mc bound can still infect, if they come into contact with CCR5-expressing cells before they undergo inactivation (See Supplementary Figure 8a of the revised manuscript).

The occupancy of the Env trimers by the CD4mc is determined by the concentration of the CD4mc and the affinity of the CD4mc for the Env trimer. The population of Env trimers in a virus preparation exhibits CD4mc occupancies (0, 1, 2 and 3 CD4mc per trimer) described by a binomial distribution (See Supplementary Figure 8b of the revised manuscript). Under the experimental conditions of the high-dose SHIV-C5 challenge, a relatively high concentration of BNM-III-170 was used to ensure that the majority of viral Env spikes had some CD4mc bound. At this BNM-III-170 concentration, we expect that the majority of the viral Env trimers will have 1

or 2 CD4mc bound and are thus potentially sensitized to antibody neutralization. A fraction of these Envs will have all three Env protomers occupied by the CD4mc and will be directly inactivated, as discussed above. Importantly, a small fraction of the viral Envs will be free of CD4mc and therefore insensitive to neutralization by the gp120-elicited antibodies in the monkey serum. The data in the revised manuscript show that the SHIVs infecting the Group-3 monkeys are resistant neither to the direct antiviral effects of BNM-III-170 nor to BNM-III-170-induced sensitization to antibody neutralization (Supplementary Figure 6b). The SHIVs infecting the Group-3 monkeys were derived stochastically from the SHIV-C5 challenge stock (Figure 4). These observations are consistent with the infecting SHIV Envs failing to bind adequate amounts of CD4mc or CD4mc + antibody before engaging a host target cell. We have modified the manuscript to present more thoroughly how this mechanistic model of CD4mc action explains the observed protection results.

As we noted in our previous reply to Reviewer 3, establishing correlates of protection would be applicable in a much larger study. In the Hudgens *et al.* study (*Biometrics*, 2009), they determined that at least 20 monkeys in each arm of a vaccine study are needed to have sufficient power to detect an immune correlate of protection. Moreover, if the viruses infecting the Group-3 monkeys did not bind BNM-III-170 and are intrinsically neutralization-resistant, there need not be any correlation between infection and host antibody responses.

We agree that repeating the study in a separate group of monkeys would allow mucosal biopsies to be performed. This is, however, beyond the scope of the present study. Mucosal biopsies will be incorporated into future studies of vaginal rings delivering CD4mc.

Reviewer 2

The authors have satisfactorily addressed many of the points raised in the first review. However, the logical and practical aspects of this vaccine-boosting concept require further investigation.

Specific points:

Point a. Authors' response: The major HIV-1 Env conformation recognized by ADCC-mediating antibodies is the CD4-bound conformation. In HIV-1-infected cells, two viral proteins, Vpu and Nef, specifically interact with the cytoplasmic tail of CD4 and remove CD4-Env complexes from the cell surface. Thus, HIV-1-infected cells are relatively resistant to ADCC (...) ADCC would

be expected to be relatively inefficient in this setting. Therefore, virus-neutralizing antibodies likely contribute to the protection observed in the Group 3 monkeys.

This point is controversial. When CD4 T cells are infected in vitro, antibodies to cluster-A epitopes, like A32, are the first and most prominently reactive with the infected cell surface. In fact, it takes several days for Vpu and Nef to fully downregulate CD4. Thus, recently infected CD4 T cells should be excellent targets for ADCC in vivo, and it remains unclear why ADCC-mediating antibodies, if protective, should not work efficiently even in the absence of CD4mc. Regarding ADCC, the authors' statement that "ADCC would be expected to be relatively inefficient in this setting" seems to contradict the emphasis given to ADCC as a potential mechanism of protection in the manuscript. Conversely, the role of virus-neutralizing antibodies in protection is somewhat difficult to envisage here since binding to the CD4 mimetic would induce the trimer to assume an open conformation and make the virus non-infectious.

Response: As discussed above in our response to Reviewer 1, the role of neutralizing antibodies in the protection observed in the Group-3 monkeys is difficult to envisage if the binding of a single CD4mc inactivates the Env trimer. Supplanting this incorrect model with the above-described model, in which the consequences for Env function depend upon the stoichiometry of CD4mc binding to the Env trimer, clarifies how CD4mc can sensitize HIV-1 to neutralization by otherwise ineffectual antibodies.

We do not mean to minimize the potential contribution of ADCC as a mechanism of protection in Env-vaccinated individuals administered CD4mc locally or systemically. Our point is that, in this particular experimental protocol, the only source of CD4mc available to the infected cell must come from the virus inoculum, and the CD4mc would be expected to be very low in concentration by the time infected cells begin to express Env. We have tried to clarify this point in the revised Discussion.

The reviewer asks why ADCC-mediating antibodies don't work efficiently in the absence of CD4mc, as shown in our Figure 2. Antibodies against cluster-A epitopes on gp120, the major targets of ADCC, do not efficiently recognize intact Env trimers on the surface of HIV-1-infected cells. This is because the cluster-A epitopes are buried within the closed Env trimer. When HIV-1 Nef or Vpu is deficient, CD4 is not downregulated, allowing exposure of A32 epitopes on Env-CD4 complexes on the cell surface. Nef is expressed early in HIV-1 infection (within 8-12

hours) and at high levels (Robert-Guroff *et al. J. Virol.* 64: 3391-9 (1990); Klotman *et al. PNAS* 88: 5011-5 (1991); Chen *et al. J. Virol.* 70: 6044-53 (1996)), allowing rapid downregulation of cell-surface CD4 (Schwartz *et al. J. Virol.* 69: 528-33 (1995)). Later, Vpu and Env are expressed from a bicistronic mRNA, ensuring that at least as many Vpu proteins as Env glycoproteins are synthesized. Vpu downregulates CD4 in the endoplasmic reticulum. In our experience, Nef and Vpu down-regulation of cell-surface CD4 in HIV-1-infected cells is extremely efficient. Thus, the intact Env trimers on the surface of infected cells do not expose cluster-A epitopes efficiently.

As the reviewer noted, cluster-A antibodies like A32 can bind cells in HIV-1-infected cultures. When we examined these cultures with an ultra-sensitive RNA-flow Fluorescence In Situ Hybridization (FISH) method, we found that the vast majority of A32-positive cells were uninfected bystander CD4-positive T cells rather than infected cells. The gp120 shed from infected cells binds CD4 on these bystander cells, exposing the A32 epitope.

In summary, the recognition of HIV-1-infected, virus-producing cells by readily elicited antibodies that mediate ADCC is inefficient. CD4mc opens the Env trimer and allows improved binding of ADCC-mediating antibodies.

Point b. Authors' response: This is a good suggestion. In addition to Supplementary Table 1, we provide Supplementary Figure 1 showing the immunization received by each monkey in the study and summarizing the results of the SHIV-C5 challenges.

This point has been satisfactorily addressed.

Point c. Authors' response: In this proof-of-principle experiment, the virus was mixed with the CD4mc to ensure contact of the infecting virus with the CD4mc. As discussed above, for real world application, sustained release of the CD4mc, either mucosal or systemic, would be employed in vaccinated individuals. Pharmacokinetic and toxicity studies of CD4mc in monkeys are ongoing, and will inform future attempts to deliver these compounds for prevention and/or treatment of HIV-1 infection.

This remains a major hurdle in the practical development of this vaccine-boosting concept.

Pharmacokinetic and toxicity/immunogenicity studies are mandatory to draw any conclusion on the potential applicability of this approach.

Response: We agree that pharmacokinetic and toxicity studies are important for practical applications of CD4mc, and the current status of these ongoing studies is summarized above.

Minor points:

The authors have satisfactorily addressed all the minor points.

Reviewer 3

The authors have adequately addressed many of this reviewer's specific comments from the original review, with some new or remaining outstanding minor points listed below. Overall, this work represents an important proof-of-principal animal protection study for a novel and interesting concept, that of sensitizing virus to vaccine-elicited antibodies using a small compound that simulates CD4 binding to HIV-1 Env. But the larger question pertaining to utility is difficult to overlook. The broader impact of this work in the field will likely be somewhat limited due to the practical challenge of applying this strategy as an HIV-1 prophylaxis, since it would require delivery of CD4mc to sites of HIV-1 exposure to treat and render virus susceptible to vaccine elicited antibodies.

Major limitations of the study related to study design also hamper its potential impact, namely: 1) the unclear effect of CD4mc alone on virus infectivity (group 1) confounds interpretation of the group 3 efficacy, 2) the vaccine regimen consisted of seven gp120 protein immunizations prior to the first challenge, and the gp120 sequence differed for each of the first four immunizations, a regimen difficult to envisage clinically, and 3) protein boosting shortly before each of the first 3 challenges. Their potential impact on the primary finding of macaque protection against SHIV challenge could be substantial, but cannot be discerned from the present study.

Response: The sustained delivery of CD4mc to mucosal sites of potential HIV-1 exposure via vaginal rings or implants is discussed above. Also as discussed above, any contribution of the

direct antiviral effects of the CD4mc alone to the protection of the Group-3 monkeys does not affect the validity of our conclusions.

Because the Group 2 and Group 3 monkeys in this study were recruited from a previous study, the gp120 immunization regimen admittedly may be more complicated than is necessary for successful application of our approach. In other studies, antibodies capable of neutralizing CD4mc-sensitized HIV-1 have been generated with a single Env immunogen. Nonetheless, future studies will be needed to determine if more simplified immunization protocols are equally protective.

The results of SHIV-C5 Challenges 4 and 5 indicate that substantial protection can be achieved without protein boosting in the immediate period before the virus challenges.

Minor points

1) The addition of gp120-specific binding antibody data in Supplemental Figures 2 and 7 adequately address the question of vaccine-elicited antibody durability. However, why do the ELISA values differ for the day of challenge #1 time point between Figures 2 and 7? Also, it would be helpful present a summary of the Figure 7 longitudinal responses in a single panel within this figure to illustrate the response decay kinetics in Group 3. The responses appear unusually stable with little to no decrease during the 6 months post-vaccination. The authors should comment on what may have contributed to this durability. The nine protein vaccinations?

Response: The ELISAs in Supplementary Figure 2 and Supplementary Figure 7 were performed with different preparations of HIV-1_{YU2} gp120. Therefore, although all of the results within each figure can be compared, there are some quantitative differences between the peak ELISA readings in the different figures. Nonetheless, the estimates of the serum antibody titers are similar in the two sets of assays.

We have added a panel showing the titer of the anti-gp120 antibody response as a function of time after boosting to Supplementary Figure 7. We agree that the anti-gp120 antibody titers are quite stable over six months. Without additional experiments, we cannot speculate on whether this longevity results from the particular gp120 immunogen, the immunization protocol, adjuvant, or other variables.

2) *It appears that Groups 2 and 3 were not entirely balanced with respect to the passively transferred antibodies administered during the first four gp120 vaccinations. For example, from Supp. Figure 1, M2850 was administered to 2 animals in G2 and 4 in G3.*

Response: As Mamu A*01 alleles have been previously shown to influence the susceptibility of rhesus macaques to SIV and SHIV challenges, our primary goal was to match the number of Mamu A*01-positive monkeys in Groups 2 and 3. This necessitated some imbalance in the number of Basilizimab-, M2850- and CH65-treated monkeys between Groups 2 and 3, particularly because there are non-even numbers of animals in two of these treatment groups. We mention this fact in the revised Online Methods.

3) *Supplemental Figure 1 is an important addition to the manuscript, and attempts to clarify many details about the history and genetic background of each animal. The authors are commended for fully disclosing these details. However, this reviewer finds the table quite complicated and extremely difficult to follow. A simple timeline summarizing the entire protocol within the Main Figure 3 would improve transparency regarding the full vaccination and challenge schedule, including indication of the ongoing vaccinations during challenges 1-3. Burying the vaccination regimen and timing relative to the challenges in Supplemental is a disservice to readers.*

Response: Other than Supplementary Figure 1, we haven't found a way to simplify the timeline while summarizing the entire protocol. Should the manuscript be accepted for publication, we will ask the editors if Supplementary Figure 1 can be accommodated within the main figures.

4) *The main text of the manuscript fails to mention that different gp120 immunogens were administered over the course of the seven pre-challenge vaccinations. This is misleading, as the current text suggests that a single gp120 immunogen was used, the CH505 transmitted founder Env, when in fact that was only used once for the first vaccination. It is an important technical detail that should be stated in the Main manuscript, and included in the vaccination timeline presented in Figure 3, as suggested above. It raises possible questions about whether multiple Env variants would be a requirement for this vaccination to be effective.*

Response: As discussed above, we agree and have added this information to the main text of the revised manuscript.

5) While a formal immune correlates analysis may not be feasible with a group size of 8, it is noteworthy that the gp120-specific titers in the Group 3 animals at the time of first challenge appear to show no association with time to infection. If true, the authors should acknowledge this observation in the Results and offer an explanation of why this is not incongruous with their proposed mechanism of protection in the Discussion.

Response: All of the Group-3 monkeys had high titers of antibodies against Env and effectively neutralized viruses with the C5 Env in the presence (but not the absence) of the CD4mc. As discussed above in our response to Reviewer 1, the viruses infecting the Group-3 monkeys may not have bound BNM-III-170 and thus could have been resistant to neutralization by the antibodies in the host animal. According to this model, the Group-3 monkeys that became infected would not necessarily have a lower titer of anti-Env antibodies than monkeys that were protected. This explanation has been added to the revised manuscript (Supplementary Figure 8 legend).

We thank the reviewers for their helpful suggestions. We feel that the revised manuscript is now significantly improved, and trust that it is suitable for publication in *Nature Communications*.

Sincerely,

Joseph Sodroski

Reviewers' comments:

Reviewer #1 (Remarks to the Author):

The latest round of responses satisfactorily addresses all my remaining concerns. I would just include a statement in the manuscript summarizing the important safety data they describe in their response.

Reviewer #3 (Remarks to the Author):

Comments for Editor & Authors:

The authors have adequately addressed many of this reviewer's specific comments from the second review.

However, this reviewer remains unsatisfied with the authors' response to Minor point #3 from the previous review:

"Supplemental Figure 1 is an important addition to the manuscript, and attempts to clarify many details about the history and genetic background of each animal. The authors are commended for fully disclosing these details. However, this reviewer finds the table quite complicated and extremely difficult to follow. A simple timeline summarizing the entire protocol within the Main Figure 3 would improve transparency regarding the full vaccination and challenge schedule, including indication of the ongoing vaccinations during challenges 1-3. Burying the vaccination regimen and timing relative to the challenges in Supplemental is a disservice to readers."

Author Response: "Other than Supplementary Figure 1, we haven't found a way to simplify the timeline while summarizing the entire protocol. Should the manuscript be accepted for publication, we will ask the editors if Supplementary Figure 1 can be accommodated within the main figures."

This reviewer feels strongly that a schematic of the full vaccination and challenge schedule be included as a panel within Figure 3. To assist the authors, I include an attachment: a hand-drawn

rough sketch of how this could be depicted in a simplified manner distinct from Supplementary Figure 1. (Supplementary Figure 1 should remain in supplemental due to its complicated presentation.) A polished version of this sketch could easily be accommodated within Figure 3. I repeat, it is a disservice to readers to not include such a schematic as it provides important context for interpreting the outcome of the SHIV challenges. This reviewer is concerned by the lack of effort made by the authors to address this point as it suggests a lack of willingness to be transparent about the study design.

Reviewer 1

The latest round of responses satisfactorily addresses all my remaining concerns. I would just include a statement in the manuscript summarizing the important safety data they describe in their response.

Response: We have included a statement in the revised Discussion summarizing the safety/toxicity results to date.

Reviewer 2

The authors have adequately addressed many of this reviewer's specific comments from the second review.

However, this reviewer remains unsatisfied with the authors' response to Minor point #3 from the previous review:

"Supplemental Figure 1 is an important addition to the manuscript, and attempts to clarify many details about the history and genetic background of each animal. The authors are commended for fully disclosing these details. However, this reviewer finds the table quite complicated and extremely difficult to follow. A simple timeline summarizing the entire protocol within the Main Figure 3 would improve transparency regarding the full vaccination and challenge schedule, including indication of the ongoing vaccinations during challenges 1-3. Burying the vaccination regimen and timing relative to the challenges in Supplemental is a disservice to readers."

Author Response: "Other than Supplementary Figure 1, we haven't found a way to simplify the timeline while summarizing the entire protocol. Should the manuscript be accepted for publication, we will ask the editors if Supplementary Figure 1 can be accommodated within the main figures."

This reviewer feels strongly that a schematic of the full vaccination and challenge schedule be included as a panel within Figure 3. To assist the authors, I include an attachment: a hand-drawn rough sketch of how this could be depicted in a simplified manner distinct from Supplementary Figure 1. (Supplementary Figure 1 should remain in supplemental due to its complicated presentation.) A polished version of this sketch could easily be accommodated within Figure 3. I repeat, it is a disservice to readers to not include such a

schematic as it provides important context for interpreting the outcome of the SHIV challenges. This reviewer is concerned by the lack of effort made by the authors to address this point as it suggests a lack of willingness to be transparent about the study design.

Response: We thank the reviewer for suggesting a simplified version of the vaccination and challenge schedule. This schematic diagram is now included in Figure 3a of the revised manuscript.

We thank the reviewers for their helpful suggestions, which have improved the manuscript. We trust that the revised manuscript is now suitable for publication in Nature Communications.